# Anti-Hypoxia Nanoplatforms for Enhanced Photosensitizer Uptake and Photodynamic Therapy Effects in Cancer Cells

**DOI:** 10.3390/ijms24032656

**Published:** 2023-01-31

**Authors:** Nkune Williams Nkune, Heidi Abrahamse

**Affiliations:** Laser Research Centre, Faculty of Health Sciences, University of Johannesburg, P.O. Box 17011, Doornfontein 2028, South Africa

**Keywords:** photodynamic therapy, nanoparticles, hypoxia, three-dimensional cell culture models

## Abstract

Photodynamic therapy (PDT) holds great promise in cancer eradication due to its target selectivity, non-invasiveness, and low systemic toxicity. However, due to the hypoxic nature of many native tumors, PDT is frequently limited in its therapeutic effect. Additionally, oxygen consumption during PDT may exacerbate the tumor’s hypoxic condition, which stimulates tumor proliferation, metastasis, and invasion, resulting in poor treatment outcomes. Therefore, various strategies have been developed to combat hypoxia in PDT, such as oxygen carriers, reactive oxygen supplements, and the modulation of tumor microenvironments. However, most PDT-related studies are still conducted on two-dimensional (2D) cell cultures, which fail to accurately reflect tissue complexity. Thus, three-dimensional (3D) cell cultures are ideal models for drug screening, disease simulation and targeted cancer therapy, since they accurately replicate the tumor tissue architecture and microenvironment. This review summarizes recent advances in the development of strategies to overcome tumor hypoxia for enhanced PDT efficiency, with a particular focus on nanoparticle-based photosensitizer (PS) delivery systems, as well as the advantages of 3D cell cultures.

## 1. Introduction

Cancer remains one of the leading causes of death around the world. The prevalence of cancer is expected to rapidly increase in the next 15–35 years due to drug resistance [1]. Conventional cancer therapies include surgical excision, chemotherapy, immunotherapy and radiation therapy, which are administered alone or in combination depending on the patient’s condition, stage, and location of the tumor [2]. However, these treatment approaches rarely yield good prognoses in patients due to their undesirable side effects on normal tissues and the development of complex resistance mechanisms [2,3].

Photodynamic therapy (PDT) is a novel therapeutic procedure that incorporates three essentials—a photosensitizer (PS), oxygen and light [1]. PDT relies on the retention of the PS in tumors followed by their light activation to trigger photooxidative reactions that damage and destroy tumors [4]. However, clinical use of conventional PSs is drastically hampered by their inherent characteristics including poor solubility, aggregation in aqueous conditions and poor tumor-targeting ability [4]. To circumvent these limitations, various nanocarrier platforms (dendrimers, micelles, liposomes, mesoporous silica, metal-organic frameworks and graphene-based nanoparticles) have been adopted to augment water solubility and enhance the bioavailability of PSs in tumor tissues [4,5]. These nanocarriers endow PSs within high affinity for tumors via passive targeting strategies, which are attributed to the enhanced permeability and retention (EPR) phenomenon of nanocarriers in tumor tissues [4]. To further enhance the bioavailability and target-selectivity of PSs, targeting moieties, typically antibodies, carbohydrates and peptides, are encapsulated into nanocarriers, which mitigates unwanted side effects by decreasing the off-target cytotoxicity of nanocarriers [1].

Oxygen is the most integral component of PDT. Nevertheless, the uncontrolled proliferation of cancer cells stimulates a hypoxic tumor microenvironment [6,7,8], which drastically counteracts PDT performance. Therefore, recent studies state that the hypoxic nature of tumor tissue is a great impediment to effective intracellular reactive oxygen species ROS) yield [9]. The demand for intracellular oxygen during the PDT photochemical processes may exacerbate tumor hypoxia, potentially resulting in decreased PDT potency [9]. To alleviate hypoxia, various chemical and engineering strategies have been identified to boost oxygen content in targeted tissues and improve the therapeutic index of PDT in cancer treatment [10]. There are several strategies that have been developed to circumvent tumor hypoxia, such as increasing oxygen supply, the disruption of tumor ECM, the inhibition of tumor O_2_ consumption and the inhibition of angiogenic factors [11,12,13].

Anticancer drugs are typically tested on 2D cell culture models, which involve growing flat monolayer cells on a glass or plastic cell culture flask with very little cell–cell and cell–matrix interaction, as seen in natural tumors (Figure 1A) [14,15]. The 2D cell culture systems are simple, convenient, and economical [15]. However, the main drawback of traditional 2D cell culture is the failure to simulate the native solid tumor architecture and microenvironments, including cellular heterogeneity, limited oxygen and nutrient distribution, cell–cell transduction, growth kinetics, cellular interactions, and resistance to therapies [16,17]. Irrelevant data obtained from 2D cell culture models often do not predict in vivo responses, and animal models are costly, labor-intensive, controversial regarding ethical issues, and their outcomes differ depending on animal species [15,18]. Such concerns prompted the development of 3D cell culture platforms, a promising tool to combat the discrepancy between in vitro studies and clinical trials [19,20]. Multicellular tumor spheroids (MCTS) are ideal 3D models for drug testing because they biologically and morphologically mimic solid tumors (Figure 1B) [19].

MCTS exhibit a layered internal cell distribution similar to that seen in solid tumors (Figure 1C) [15,21]. This is due to mass transportation restrictions that obstruct the diffusion of oxygen, nutrients, and metabolic wastes through the spheroid, resulting in distinct gradients [21]. Due to their sufficient exposure to oxygen and nutrients, highly proliferating cells are found in the external layer of spheroids and closely resemble cancer cells nearer to the blood vessels in vivo [15,21]. Cells in the middle layer of the spheroid are normally quiescent because cellular metabolic activity drops gradually with increasing distance from the spheroid outer rim [14,15]. Lastly, the spheroid core is characterized by necrotic cells due to limited oxygen and nutrient gradients as well as the accumulation of waste products [21].

Herein, we investigate nanoparticle-based PS delivery systems incorporated with anti-hypoxia agents to improve the PS bioavailability and attenuate hypoxia in 3D tumor spheroids for enhanced PDT efficacy and better treatment outcomes.

## 2. Photodynamic Therapy (PDT) and Photosensitizers (PSs)

PDT is a phototherapeutic procedure used to eradicate various oncological and non-oncological disorders [22]. PDT exhibits several advantages over standard cancer treatments, which include improved tumor affinity, negligible systemic toxicity and the availability of continuous treatments [23]. It relies on three fundamentals: (1) a photosensitizing molecule or PS accumulated in tumor tissues; (2) an appropriate wavelength of light to activate the PS; and (3) molecular oxygen in tumor tissues [22,24]. Upon photoactivation of a PS through illumination with a suitable wavelength, it reacts with surrounding molecular oxygen and triggers photochemical reactions that generate reactive oxygen species (ROS) and singlet oxygen that render cell death [3].

PSs are typically classified as first-, second-, and third-generation according to the time of invention and their unique characteristics [25]. Ideally, a PS should exhibit precise tumor targeting, negligible dark toxicity, maximum absorption of light at wavelengths from 600 to 800 nm, a high degree of chemical purity and stability, high quantum ROS yields, simple synthesis and rapid elimination from healthy tissues [5,24]. First-generation PSs, such as hematoporphyrin derivative (HpD) and photofrin, were the first PSs to be explored in PDT applications [22]. Despite their extensive application in PDT, these PSs are limited by poor tumor targeting abilities, low chemical purity, dark cytotoxicity, poor absorption and tissue penetration of light due to their strong absorption at short wavelengths (>650 nm), as well as their delayed elimination from healthy tissues [5,24,25]. These limitations led to the development of second-generation PSs [25].

Second-generation PSs consist of hematoporphyrin-derived and synthetic PSs such as 5-aminolevulinic acid, benzoporphyrin derivatives, texaphyrins, thiopurine derivatives, chlorin, as well as bacteriochlorin analogues and phthalocyanines [24]. These PSs provide several advantages over first-generation PSs, such as higher chemical purity, improved singlet oxygen quantum yield and deeper penetration to targeted tumor sites due to their maximum absorption within the therapeutic window [5,24]. In addition, they exhibit fewer side effects, which is attributed to a higher affinity for tumor tissues and rapid elimination of the PS from the body [5]. However, the main obstacle for second-generation PSs is their hydrophobic nature, which significantly hampers their clinical application and calls for the development of novel methods of PS delivery [1]. To circumvent this challenge, third-generation PSs have been extensively investigated [5].

Third-generation PSs are second-generation PSs modified with targeting moieties, such as antibodies, peptides, and carbohydrates, or incorporated with biological conjugates such as nanoparticles, liposomes and micelles to enhance their physical and chemical properties and their accumulation at targeted tumor regions [1,5,25]. The main purposes of third-generation PSs are to reduce the dosage and minimize unwanted side effects on healthy cells while actively targeting tumor tissues, resulting in improved selectivity [1,25].

## 3. Photodynamic Therapy (PDT) Mechanism of Action and Cell Death

PDT damages and obliterates cancer cells via two mechanisms of action (type I and type II), which both rely on oxygen molecules in targeted tumor sites (Figure 2) [24]. Upon illumination of a PS with a wavelength of light corresponding to its absorption spectrum, the PS molecules transform from a ground state into a singlet excited state [22]. The excited singlet PS loses its excess energy either through fluorescence or heat production [26]. However, the excited singlet PS may undergo intersystem system crossing, a process whereby the spin of its excited electrons inverts to form a more stable excited triplet state [27]. In the type I mechanism, the triplet excited state PS transfers energy to biomolecules within a tumor to form radicals, which react with molecular oxygen to generate ROS, such as hydrogen peroxide, superoxide anion radicals, and hydroxyl radicals [22]. In the type II mechanism, the excited triplet state PS transfers energy directly to ground-state oxygen (^3^O_2_) to form extremely oxidizing singlet oxygen (^1^O_2_) [25]. The resulting ROS and ^1^O_2_ cause tumor cell death primarily via apoptosis, necrosis, autophagy pathways, and vasculature damage, as well as triggering cytotoxic immunogenic reactions [28]. The degree of photocytotoxicity is determined by several factors, including PS intracellular localization, cellular uptake rate, PS physicochemical characteristics, tumor molecular oxygen levels, and light intensity [5].

Apoptosis is the most predominant mode of cell death triggered by PDT-induced photodamage to the mitochondria through the activation of various signaling pathways involving caspases, Bcl-2 proteins and proapoptotic proteins [29]. Malignant cells succumbing to apoptosis exhibit cell nuclear and membrane fragmentation [3]. Autophagy (self-eating) is a crucial cellular mechanism, which eradicates unwanted or dysfunctional components [30,31]. In cancer cells, autophagy may regulate cell growth. It has also been highlighted that phototoxicity may trigger autophagy [30,31]. As PDT elevates ROS levels, cellular stress intensifies, and Beclin-1 protein initiates autophagic pathway [30,31]. PDT was proposed to evoke autophagy by inactivating negative regulators of autophagy (e.g., mTOR, Bcl-2) instead of activating autophagic proteins (e.g., Atg7, Beclin 1, Atg5) [32,33,34]. Studies demonstrate that autophagy may exhibit cell killing or survival effects depending on the PS applied in PDT as well as the type of cell line or light dose [31]. Recent studies by Song et al. [35] revealed that photoactivated meta-tetrahydroxyphenylchlorin (m-THPC) and verteporfin (VP) induced autophagic cell death in colorectal cancer cells (HCT116 and SW480) via activation of the ROS/JNK signaling pathway.

Necrosis is an uncontrolled mode of cell death resulting from cell membrane disruption and loss of integrity [22]. Necrosis is distinguished by cytoplasmic swelling, extensive organelle annihilation, and plasma membrane destruction, which result in intracellular contents release and cancer cell decomposition [3].

The cytotoxic ROS can disrupt tumor vasculature, resulting in a loss of oxygen and nutrients to tumor tissues, ultimately leading to tumor destruction [28]. PDT can also elicit inflammatory reactions, which cause leukocyte accumulation and the activation of pro-inflammatory factors and cytokines at the tumor site [28]. Moreover, ROS can exacerbate tumor tissue damage by inducing immunogenic cell death [20]. Photodamaged cancer cells release damage-associated molecular patterns (DAMPs), which serve as danger signals [28]. These DAMPs are recognized by the innate immune phagocytes (macrophages, neutrophils and dendritic cells), which eradicate the tumor cellular debris [20,28]. Furthermore, antigen-presenting cells (APCs), particularly dendritic cells, can recognize DAMPs alone or together with tumor antigens, activating T and B-cells, which results in long-term adaptive antitumor immunity [28].

## 4. Passive and Active Intracellular Accumulation of Photosensitizers

The accumulation of PSs within a tumor can occur either passively or actively [22]. Passive intracellular accumulation is facilitated by the EPR effect attributed to the impaired lymphatic drainage within the tumorous region, which allows PSs to easily penetrate through the leaky tumor vasculature [5,22]. The EPR effect is a naturally occurring process that takes advantage of the anatomical and pathophysiological differences between tumorous tissue and normal cells to increase PS passivation in tumor cells [22]. When nanoparticles (NPs) are combined with PSs, they significantly promote the passive uptake of PSs via this EPR effect [36]. In addition, modifying PSs with NPs can enhance either the stability or solubility, as well as reduce dark toxicity and enhance localized delivery, to improve PDT treatment efficacy and minimize undesirable side effects [22]. Moreover, the small dimensions of NPs not only allow PSs to accumulate in cancer cells via passive or active targeting, but also allow these nanocarriers to mimic biological molecules and, thus, easily evade immune system barriers [22,36]. Active intracellular accumulation requires the immobilization of targeting moieties such as antibodies, folic acid (FA), peptides, small ligands and carbohydrates on the surface of PS-nanocarrier platforms, which have a high affinity for receptors overexpressed by cancer cells and so facilitate active cellular PS uptake in tumor sites [3].

Active nanocarrier systems, as opposed to passive systems, effectively facilitate the selective accumulation of PS at targeted regions only, increasing PDT efficiency while minimizing off-target toxicity and undesirable side effects on normal tissues (Figure 3) [37].

## 5. Hypoxia-Mediated PDT Resistance

Hypoxia is an inherent feature of solid tumors, which promotes various abnormal physiological alterations in the tumor microenvironment, such as angiogenesis, rapid tumor cell proliferation, and impaired lymphatic systems [10]. In this case, cancer cells obtain more energy from various metabolic pathways by consuming endogenous O_2_, particularly that from the oxidative phosphorylation (OXPHOS) metabolic pathway. Therefore, oxygen-dependent PDT inherently struggles to attain progressive ROS production in tumor tissue [38,39,40]. In fact, owing to insufficient oxygen supply, few ROS are generated in the cell following several repeated PDT episodes [10]. The increased oxygen consumption during PDT can further exacerbate hypoxia in cancer cells, and drastically hamper therapeutic outcomes by altering the proteome and genome of cancer cells (Figure 4) [10]. Additionally, hypoxia serves as a barrier to antitumor agents by manipulating apoptotic factors, autophagy, drug efflux and other signaling mechanisms [41]. It promotes survival and tumorigenesis by enabling cells to surmount nutrient deprivation or evade their hostile environment [10,38,41]. According to research, hypoxia-inducible factor-1 (HIF-1) activates the multidrug resistance 1 (MDR1) gene in response to hypoxia. MDR1 encodes the membrane-resident P-glycoprotein (P-gp), a member of the ABC transporter family that acts as a drug efflux pump to reduce the intracellular concentration of some anticancer drugs upon administration [42,43]. Furthermore, HIF-1 is also implicated in autophagy due to its involvement in the initiation and progression of autophagosome production, including B-cell lymphoma 2 (Bcl-2), adenovirus E1B 19 kDa-interacting protein 3 (BNIP3), Beclin, BNIP3-like (BNIP3L)/NIX, phosphatidylinositol 3 kinase catalytic subunit type 3 (P13KC3), autophagy-related gene 7 (ATG7), ATG5, and ATG9A [44,45,46,47,48,49,50]. Mammalian target of target rapamycin complex 1 (mTOR C1) is a serine/threonine kinase that regulates cell growth [51]. mTOR activity is counteracted under deprivation or hypoxia, which plays a pivotal role in autophagy induction [51,52].

Therefore, hypoxia not only inhibits photochemical reactions by limiting ROS generation but also counteracts PDT-induced photodamage by activating a series of pro-survival responses, which play a key role in PDT resistance [10]. Given that oxygen supply and consumption imbalance are closely related to tumor hypoxia, several approaches have been used to increase the intra-tumoral oxygen gradient [41,53].

## 6. Current Approaches for Circumventing Hypoxia in PDT

### 6.1. Increasing Oxygen Supply

#### Hyperbaric Oxygenation (HBO)

To circumvent the hypoxic TME during PDT, various oxygen-generating nanoparticles have been developed, which often capitalize on the oxidative stress in the tumor to decompose endogenic hydrogen peroxide (H_2_O_2_) to increase oxygen levels inside the tumor [54]. HBO therapy can alleviate tumor hypoxia in several ways. One approach is to inject additional O_2_ in targeted regions to be treated with ^1^O_2_ itself, H_2_O_2_ or other molecules. Another approach is to promote the generation of ROS at the expense of ^1^O_2_ by converting the photochemistry of PSs from a type II to a type I mechanism [53].

Studies by Mei et al. 2019 investigated the in vitro phototoxicity of 5-aminolevulinic acid (5-ALA) with and without HBO on 2D monolayers of human squamous carcinoma (A431 cell line) [55]. Before the addition of 5-ALA, the cells were flushed with a mixture of 98% O_2_ and 2% CO_2_ (0.25 MPa) hourly per day for HBO assays [55]. The results indicated that the incorporation of HBO with PDT synergistically decreased A431 cell proliferation. Furthermore, Mei and colleagues also noted that this combined therapy markedly induced apoptosis and autophagy.

Studies conducted by Chen et al. [56] explored hyperoxygenation in in vivo mammary carcinoma tumors inoculated in C_3_H mice (6–8 mm). Tumors received photofrin and laser irradiation at 630 nm (200 J/cm^2^, 150 mW/cm^2^). Hyperbaric conditions were initiated by subjecting the mice to an atmospheric pressure of 3. Normal conditions (1 atm and normoxia) resulted in an approximately 20% reduction in tumor mass after 60 days. In normobaric conditions (NBO, 100% O_2_), tumor reduction increased to 80%, whereas in hyperbaric conditions (HBO, 100% O_2_), a 60% reduction in tumor mass was noted [56]. Furthermore, the reduction in light fluency to 75 mW/cm^2^ combined with NBO exerted 70% tumor reduction_._ Chen and colleagues noted that the degree of phototoxicity in all cases (HBO and NBO) doubled, compared to normal atmospheric conditions [56].

Recent studies by Li et al. [54] investigated the phototoxic effect of an upconversion PS nanocarrier system coupled with hyperbaric conditions. They conjugated upconversion nanoparticles (UCNPs) with Rose Bengal PS (UCNP-PS) and combined them with HBO to disrupt the extracellular matrix for improved PDT treatment upon excitation at 808 nm [54]. It was observed that UNPS-mediated PDT and HBO significantly decomposed collagen fibers in the extracellular matrix and drastically reduced hypoxia, as well as improving the bioavailability of the PS in tumor vasculature [54].

Studies by Hjelde et al. [57] investigated the effect of hyperoxia on PDT and lipid peroxidation on human colon cancer monolayer cells, SW480, WiDr and one rat bladder cancer cells, AY-27. The cells were incubated with 2 mM 5-ALA- for 3.5 h and subjected to laser irradiation at 435 nm before exposure to hyperoxia (100, 200, 300, and 400 kPa). Hjelde and colleagues revealed that PDT conducted under normoxia triggered peroxidation and caused a significant decrease in cell viability [57]. However, the exposure to hyperoxia had insignificant effect on the decrease in cell viability.

Maier et al. [58] conducted a clinical pilot study on patients diagnosed with advanced esophageal cancer to assess the influence of hyperbaric oxygen on PDT. The patients were treated with hematoporphyrin derivative (HpD) (2 mg/kg). Forty-eight hours post treatment, the patients were exposed to laser irradiation at 630 nm and fluence of 300 J/cm^2^. Of the participants, 14 (12 diagnosed with stage III cancers and 2 with stage IV cancers) received PDT alone, and 17 participants (15 with stage III cancer and 2 with stage IV cancers) received PDT combined with HBO at an absolute atmospheric pressure (ATA) of 2. Transcutaneous PO_2_ levels of 500–700 mmHg under HBO, versus transcutaneous PO_2_ levels of 60–75 mmHg under normobaric conditions, were assessed. The study noted a significant decrease in tumor volumes in both groups, with the PDT/HBO group showing a favorable response [58]. According to Maier et al. [58], the 1-year survival rate for PDT groups was 28.6% versus 41.2% for the PDT HBO group, indicating that HBO enhanced the efficacy of PDT.

Schouwink et al. [59] investigated the effect of increasing the oxygenation levels of tumors during PDT. Human malignant mesothelioma (H-MESO1) xenograft models were treated with 0.15 or 0.3 mg/kg of *meta*-tetrahydroxyphenylchlorin (*m*THPC) and after intervals of 24–120 h, the tumors were subjected to laser irradiation at 652 nm using fluence rates of 20, 100 or 200 mW/cm^2^. To enhance tumor oxygenation and phototoxicity, nicotinamide (300 mg/kg) and/or carbogen (mixture of 5% CO_2_ and 95% O_2_) were administered 30 min and/or 5 min prior to irradiation, respectively. The researchers observed that the survival rate 24 h after PDT was 11% for PDT alone and 58% for PDT and carbogen, with nicotinamide having no effect [42]. Table 1 summarizes studies that have used HBO for O_2_ supply in PDT.

### 6.2. Improving Oxygen Circulation

#### 6.2.1. Hemoglobin (Hb) and Red Blood Cells (RBCs)

Endogenous oxygen binds to hemoglobin (Hb) in red blood cells (RBCs) and is transported to various organs and tissues in physiological conditions [41]. A study by Lou et al. [60] developed a biomimetic lipid–polymer nanoparticle composed of synthetic RBCs, indocyanine green (ICG) and Hb, denoted as I-ARCs. The study was conducted on the 2D monolayers of MCF-7 breast cancer cells and subjected to a light dose (100 mW/cm^2^, 808 nm, 5 min). I-ARCs were compared to deoxy-I-ARCs and ICG nanoconjugate (INPs). Lou and colleagues reported that I-ARCs produced more ROS (9.5 times more than INPs) and formed 10.7 times more ferryl-Hb than deoxy-I-ARCs [60]. Furthermore, I-ARCs exerted the best cell-killing effects on MCF-7, reducing cell viability to 8.9% [60]. Similar observations were obtained in PDT-treated MCF-7 xenograft tumor models when comparing I-ARCs to other treatments [60].

Two years later, Liu and colleagues [61] investigated the phototoxic effect and O_2-_carrying ability of Hb-nanoconjugated biodegradable polypeptides, Hb-BODIPY-Br2 NPs, denoted as p-Hb-B-NPs, on HepG2 monolayers (620 nm, 25 mW/cm^2^, 10 min) under normal or N_2_ atmosphere and compared to that of p-B-NPs. Both nanoconjugates exerted a cytotoxic effect under a normal atmosphere, with the best effect achieved at a concentration of 4 µM p-Hb-B-NPs; however, an insignificant effect was noted under an N_2_ atmosphere [61]. Additionally, it was noted that p-Hb-B-NPs could release oxygen even in hypoxic conditions. A study conducted by Chen et al. [62] synthesized a bioinspired hybrid protein oxygen nanocarrier incorporated with Chlorin e6 (Ce6) (C@HPOC) via intermolecular disulfide bonds for oxygen-augmented immunogenic PDT. The ^1^O_2_ yield of C@HPOC was investigated in in vitro-cultured 4T1 breast cancer cells and compared to Ce6 and C@HSA. It was noted that C@HPOC generated the most significant ^1^O_2_ levels compared to the other systems upon laser irradiation (600 nm, 0.1 W/cm^2^). Ce6-mediated PDT using C@HPOC at a concentration of 1 μg/mL resulted in an 80% increase in the apoptotic cell population compared to Ce6 and C@HSA under the same conditions, suggesting that C@HPOC enhanced the efficiency of PDT. Moreover, in vivo results exhibited the remarkable oxygen-boosted PDT abilities of C@HPOC and the elicitation of systemic antitumor immune responses that could destroy primary tumors and effectively inhibit metastasis [62].

Tang et al. [63] developed a novel red blood cell (RBC)-mediated PDT, or RBC-PDT to alleviate hypoxia. Tang and colleagues conjugated phthalocyanine ZnF_16_-Pc to ferritin NPs and then incorporated the ZnF_16_-Pc-loaded ferritins (P-FRT) onto a RBC membrane to form P-FRT-RBC-NPs. In vitro investigations revealed that P-FRT-RBC-NPs could rapidly generate ^1^O_2_ even under low-oxygen conditions. In vivo, P-FRT-RBCs were administered to U87MG human glioma tumor-bearing mice and subjected to laser irradiation at 671 nm (100 mW/cm^2^_,_ 30 min). P-FRT-RBC demonstrated significant phototoxicity with a remarkable tumor suppression (76.7%) due to the co-delivery of the O_2_ and PS [63].

Studies by Wang et al. [64] developed a novel approach to evade biological barriers and selectively deliver O_2_ in the hypoxic region under near-infrared (NIR) control. To create RBC microcarriers, NIR orthogonal UCNPs were modified with a novel ultrasensitive specific hypoxia probe (HP) and Rose Bengal (RB) attached to the surface of RBCs. In vitro, upon exposure to low-oxygen conditions, the HP attached to the RBC microcarriers could be converted into an active state to facilitate the O_2_ release from oxygenated Hb following photoactivation at 980 nm. Phototoxic effects increased rapidly under 808 nm photoactivation, owing to the sustainable O_2_ yield from RBC microcarriers. Consequently, the highest photodamage (60%) was noted after alternately subjecting RBC microcarriers to 980 nm and 808 nm laser irradiation. Similar observations were obtained in U87MG solid tumor-bearing mice, showing significant tumor regression upon alternating 980 nm and 808 nm laser irradiation [64].

Cao et al. [65] assembled a multi-functional nanocomposite (BP@RB-Hb) consisting of bis(pyrene) (BP), RB, Hb and nanoliposomes to enhance PDT penetration depth and antitumor efficacy. In vitro-cultured MCF-7 monolayers treated with BP@RB-Hb showed severe phototoxicity upon laser irradiation at 480 nm (150 W), which was attributed to the increased ROS yield of BP@RB-Hb. Finally, in vivo PDT studies conducted on mouse cancer models demonstrated a significant antitumor effect following the treated of the mouse with BP@RB-Hb and laser irradiation at 800 nm (390 mW/cm^2^) compared BP@RB.

Xu and colleagues [66] synthesized a novel hemoglobin-polymer conjugate (HbTcMs) to deliver both O_2_ and the PS for enhanced PDT outcomes. A well-defined amphiphilic triblock copolymer, poly(ethylene glycol) methyl ether-block-poly acrylic acid-block-polystyrene (mPEG-b-PAA-b-PS), was fabricated via atom transfer radical polymerization (ATRP). The PS, 5,10,15,20-tetra (4-carboxyphenyl) porphyrin (TCPP), and the copolymer were both covalently bound to Hb, forming a dual O_2_ and PS nanocarrier system. HbTcMs showed an increase in resistance to oxidation and digestion while maintaining their O_2_ binding affinity compared with free Hb. HbTcM nanoconjugates showed negligible dark toxicity and enhanced internalization by 4T1 cells. Furthermore, when exposed to a 600 nm laser irradiation (70 mW/cm^2^, 2 min), HbTcMs could effectively generate ^1^O_2_ and exert greater photodamage due to the availability of O_2_ supply by Hb [66].

A study by Gao et al. [67] developed a zirconium (IV)-based metal-organic framework (MOF) (UiO-66) nanocarrier for O_2_ and indocyanine green (ICG) coated with RBC membranes. Upon laser irradiation at 808 nm, the ^1^O_2_ and photothermal properties of ICG could decompose RBC membranes and facilitate the release of O_2_ from UiO-66, thereby significantly enhancing the PDT effects on hypoxic tumors. Upon 808 nm laser irradiation of MCF-7 MCTSs treated with O_2_@UiO-66@ICG@RBC, a significant increase in the number of dead cells and ^1^O_2_ yield was observed. In vivo, O_2_@UiO-66@ICG@RBC and laser irradiation at 808 nm (0.06 W/cm^2^, 1 min) resulted in a considerable inhibition of tumor growth in tumor-bearing mice. Table 2 lists studies exploring the use of RBC for O_2_ supply.

#### 6.2.2. Perfluorochemicals

Perfluorocarbons (PFCs) and hemoglobin are two fundamental types of oxygen transport agents that have been successfully used to deliver oxygen to hypoxic tissues. They are also referred to as artificial blood substitutes [41]. PFCs can dissolve large amounts of gases at high oxygen concentrations and release oxygen at low oxygen concentrations [40]. Chen et al. [68] developed an oxygen self-sufficient photodynamic therapy (Oxy-PDT) by loading perfluorohexane (PFH) nanodroplets functionalized with a lipid shell incorporating IR780 PS (diameter 200 nm). MCF-7 or CT26 murine colon adenocarcinoma cells treated with the nanoconjugate and then exposed to an 808 nm laser (2 W/cm^2^ for 20 s) showed an increased ^1^O_2_ yield compared to PFH and IR780 alone. The phototoxic effect of the nanoconjugate was noted on the two cell lines, whereas the photothermal effect was negligible. Furthermore, the phototoxic effect of the nanoconjugate was tested on CT26 cells subjected to hypoxic conditions. Cheng and colleagues noted that the nanoconjuate still had greater cytotoxicity than conventional PDT after irradiation. In vivo analysis of ^1^O_2_ production in CT26 tumor-bearing mice using singlet oxygen sensor green (SOSG) revealed that the nanoconjugate produced more ^1^O_2_ than PFH and IR780 alone. Intra-tumoral injection of the nanoconjugate and laser irradiation (808 nm, 2 W/cm^2^ for two consecutive 10 s with a 1 min interval in between) significantly inhibited tumor growth, whereas conventional PDT had no effect. In addition, the nanoconjugate was administered intravenously and irradiated after 24 h. Cheng et al. [68] noted that the tumor volume of mice treated with the nanoconjuate and laser irradiation (808 nm, 2 W/cm^2^ for 10 + 10 s) was approximately fourfold lower than in mice treated with IR780-mediated PDT alone.

Yuan et al. [69] conjugated BODIPY-Br_2_ onto the oxygen-generating fluorinated polypeptide micelles. The fluorinated nanoconjugate demonstrated a higher oxygen content than the non-fluorinated one in hepatocellular cancer HepG2 in vivo models. Similar observations were established on the ^1^O_2_ generated by the fluorinated nanoconjugate upon irradiation at 635 nm (0.5 mW/cm^2^, 360 s) versus the non-fluorinated one, supporting the role of PFCs’ oxygen supply [69]. Furthermore, with the same irradiance (635 nm, 34 mW/cm^2^, 10 min), the nanoconjugate showed a greater efficacy on HepG2 than the non-fluorinated one.

Day et al. [70] reported that fluorous porphyrin encapsulated inside PFC nanoemulsions allowed for the simultaneous delivery of O_2_ and PS to A375 cells. Significant cell death and increased ^1^O_2_ generation were observed after 420 nm laser irradiation (8.5 mW/cm^2^, 30 min), whereas an emulsion containing fluorous rhodamine exerted negligible photodamage. Sheng and colleagues [71] developed perfluorooctyl bromide (PFOB) and combined it with ICG encapsulated in nanoliposomes (LIP-PFOB-ICG). Within in vitro-cultured MDA-MB-231 breast cancer cells, LIP-PFOB-ICG exhibited the highest ROS yield in both normal and hypoxic conditions after laser irradiation at 808 nm (1 W/cm^2^ for 3 min) when compared to free ICG and LIP-ICG. Under the same conditions, cell viability decreased to 10% due to the combination of PDT and PTT effects. Mice bearing MDA-MB-231 tumors treated with LIP-PFOB-ICG and laser irradiation at 808 nm (1 W/cm^2^ for 10 min) almost obliterated the tumors beyond recovery.

Tang et al. [72] devised PFC nanodroplets combined with IR dye 800 CW (for in vivo imaging), as well as ZnF16Pc serving as a PS and perylene diimide (PDI) as a photoacoustic agent (PA) imaging agent. PFCs’ ability to carry O_2_ enhanced PDT effects in both normoxic and hypoxic conditions. Upon irradiation, the PDI agent could absorb light and convert it into heat for contrast-enhanced ultrasound imaging and for photothermal destruction. The PA intensity was captured proportionally to the PDI concentration. The phototoxic effect of the nanoconjugate on in vitro-cultured U87MG cancer cells resulted in a cell viability of less than 34% without any photothermal effect upon irradiation at 671 nm (100 mW/cm^2^ for 200 s). In vivo, the nanoconjugate obliterated tumors after subjecting the mice to 671 nm laser irradiation at 500 mW/cm^2^ for 10 min.

Tang et al. [73] investigated the phototoxicity of IR780 conjugated to perfluorohexane nanodroplets (LIP(PFH + IR780)). Upon irradiation (808 nm, 2 W/cm^2^, 20 s) of the nanoconjugate administered to CT26 colon cancer monolayers, the singlet oxygen sensor green probe (SOSG) revealed that the ^1^O_2_ yield was higher with (LIP(PFH + IR780)) than lipidic NPs without PFH, even when the nanoconjugate was in hypoxic conditions. The phototoxicity of the nanoconjugate in hypoxia demonstrated a high mortality in the CT26 cells, while lipidic NPs had no cell-killing effects. In mice bearing CT26 subcutaneous tumors, the nanoconjugate and 800 nm laser irradiation at 2 W/cm^2^ for 20 s resulted in a significant inhibitory effect on tumor growth, whereas lipidic NPs had an insignificant effect. Several studies have used PFCs as oxygen supply agents to improve PDT outcomes (Table 3).

### 6.3. In-Situ Oxygen Source

The in situ oxygen source in the tumor through the catalyst serves as an effective means to overcome hypoxia [40]. As a result of the abnormal metabolism of cancer cells, it is known that cancer cells have higher levels of H_2_O_2_ than normal cells [53,74]. Therefore, decomposing H_2_O_2_ into oxygen is the most widely used method [41]. Several metal ions or metal oxides, such as manganese oxide, heavy metals, and transition metal ions, can catalyze the generation of oxygen from excess H_2_O_2_ in tumors [41,53]. This method not only decomposes H_2_O_2_ but also reduces the cellular proliferation of cancer [53]. Studies by Xu et al. [75] developed a photoactive Mn(II) complex of boradiazaindacene derivatives (Mn1) to generate O_2_ upon irradiation (green LED, 500–600 nm, 10 W) in aqueous conditions. Mn1 was conjugated to graphene oxide (denoted as Mn1@GO), and the phototoxicity of Mn1 and Mn1@GO was evaluated on HepG-2 cells in normal and hypoxic conditions [75]. Xu and colleagues noted that Mn1 and Mn1@GO both exerted anticancer effects on HepG-2 cells following laser irradiation (green LED, 500–600 nm, 15 min). It was also noted that Mn1@GO exhibited high antiproliferative effects on hypoxic HepG-2 cells when compared to Mn1 due to its ability to react H_2_O_2_ to form active species [75].

Zhu et al. [76] synthesized a multicomponent nanocarrier system composed of chlorine e6 (Ce6) functionalized with PEGylated MnO_2_ NPS (Ce6@MnO_2_-PEG) to improve PS cellular uptake. In in vitro-cultured 4T1 cells under hypoxic conditions, Ce6@MnO_2_-PEG nanoconjugate improved the potency of light-induced PDT (661 nm, 5 mW/cm^2^, 30 min) due to the increased intracellular O_2_ level attributed to the reaction between MnO_2_ and H_2_O_2_ [76]_._ Furthermore, in vivo PDT (661 nm, 5 mW/cm^2^, 1 h) with Ce6@MnO_2_-PEG demonstrated an enhanced intra-tumoral accumulation and significantly inhibited tumor growth in mice compared to unbound Ce6. Studies by Ai et al. [77] investigated the phototoxicity of a Ce6-loaded upconversion (UCN) nanoconjugate integrated with manganese dioxide (MnO_2_) nanosheets and active-targeting hyaluronic acid (HA) biopolymer (UCNS-MnO_2_-Ce6-HA) for enhanced PDT efficacy and attenuation of hypoxia. The photoactivation of UCNS-MnO_2_-Ce6-HA using NIR 808 nm (0.4 W/cm^2^, 60 min) generated a high level of O_2_ by means of the reaction of MnO_2_ with H_2_O_2_ in the microenvironment, which resulted in an increased production of ^1^O_2_ in hypoxic conditions. In vitro-cultured murine melanoma cells (B16F10) revealed a 49% increase in cell death after receiving UCNS-MnO_2_-Ce6-HA-mediated PDT (808 nm, 0.4 W/cm^2^, 60 min) in hypoxic conditions [78].

A study by Lin et al. [78] developed an O_2_ self-sufficient active nanobioconjugate consisting of human serum albumin (HSA), MnO_2_, and Ce6 (HSA-MnO_2_-Ce6 NPs) to alleviate tumor hypoxia and thus enhance PDT treatment of bladder cancer. HSA-MnO_2_-Ce6 NPs effectively generated O_2_ upon reaction with endogenous H_2_O_2_ in vitro. Moreover, HSA-MnO_2_-Ce6 NPs achieved a two-fold increase in ^1^O_2_ yield under 660 nm laser irradiation compared with HSA-Ce6 NPs in the presence of H_2_O_2_. In vitro cell viability assays revealed that HSA-MnO_2_-Ce6 NPs themselves were not toxic but exerted inhibitory effects on bladder cancer cells (MB-49) when exposed to laser irradiation (660 nm, 5 mW/cm^2^, 15 min). O_2_ concentration in orthotopic bladder cancer increased by 3.5 times following the administration of HSA-MnO_2_-Ce6 NPs compared to pre-injection. Given the remarkable tumor-targeting ability and minimal toxicity, HSA-MnO_2_-Ce6 NPs were used to eradicate the orthotopic bladder upon photoactivation (660 nm, 200 mW/cm^2^, 15 min). The PDT with HSA-MnO_2_-Ce6 NPs showed markedly enhanced therapeutic efficacy and significantly improved the survival rate of mice compared with controls.

Chudal et al. [79] encapsulated protoporphyrin IX (PPIX) in the liposome bilayer (PPIX-Lipo), which was then integrated with MnO_2_ to form PPIX-Lipo-MnO_2_ (PPIX-Lipo-M) in order to ameliorate hypoxia for enhanced PDT treatment in breast cancer. Photoactivated PPIX-Lipo-M (365 nm, 5 min) induced more cytotoxicity on MCF-7 breast cancer cells than PPIX-Lipo when subjected to hypoxic conditions, inferring that the generation of O_2_ in hypoxic conditions enhanced the therapeutic efficacy of PDT. The hydrophilic nature of liposomes significantly augmented the solubility of PPIX. Consequently, the cellular uptake of both PPIX-Lipo and PPIX-Lipo-M increased significantly versus that of free PPIX. Overall, PPIX-Lipo-M has the ability to act as an anticancer agent that could ameliorate hypoxia and so enhance PDT efficacy.

Ji et al. [80] designed a novel multifunctional nanosystem of CaO_2_/MnO_2_@polydopamine (PDA)-methylene blue (MB) nanosheet (CMP-MB). CMP-MB demonstrated a 34.63% cellular uptake rate in cervical cancer cells (Hela). Cell viability assays showed that CMP-MB had negligible dark toxicity while destroying HeLa cells. In conclusion, the nanosystem had the ability to self-produce O_2_, which addressed the issue of tumor hypoxia and enhanced the PDT outcome.

Chu et al. [81] designed a multifunctional nanoplatform capable of imaging (fluorescence, MRI, and PA imaging), as well as a combination therapy of phototherapy (PTT and PDT). MnO_2_ nanosheets were an efficient cargo for sinoporphyrin sodium (DVDMS) PS, as well as an in situ O_2_ and nanoDVD generator, enhancing the theranostic capability of Mn^2+^-assisted assembly of DVDMS (MnO_2_/DVDMS). In MCF-7 cells and xenograft tumors, MnO_2_ and DVDMS could be decreased by glutathione (GSH) and H_2_O_2,_ which triggered the release of Mn^2+^, DVDMS, and O_2_. Moreover, the consumption of GSH, the generation of O_2_, and the formation of nanoDVD demonstrated synergy with PTT to enhance antitumor efficacy in vitro (630 nm, 75 mW/cm^2^_,_ 5 min) and in vivo (630 nm, 300 mW/cm^2^, 8 min) in MCF-7.

Wang et al. [82] designed an intelligent and biocompatible pH/H_2_O_2_-responsive PS nanocarrier composed of bovine serum albumin (BSA), Ce6, and silicon quantum dots (Si QDs) coated with MnO_2_, denoted as BCSM NPs. Under the same conditions as TME (H_2_O_2_, pH of 6.5), the novel nanosystem showed an increased production of O_2_ and a cytotoxic ^1^O_2_ yield higher than that of free Ce6_._ In vitro-cultured HeLa cells treated with BCSM NPs and 638 nm laser irradiation (5 mW/cm^2^, 30 min) noted a considerable decrease in cell viability (20%) and a high production of cytotoxic ^1^O_2_. BCSM NP-mediated PDT (638 nm, 5 mW/cm^2^, 5 min) significantly inhibited the growth of HeLa cells harbored in mice. Several studies have used different catalysts to decompose endogenous H_2_O_2_ in order to generate O_2_ in the tumor tissue (Table 4)

### 6.4. Disruption of Tumor Extracellular Matrix (ECM)

ECM is an essential constituent of the tumor microenvironment which plays a pivotal role in tumor initiation, development, metastasis and drug resistance [83]. Proteins such as glycoproteins, elastin, collagen and proteoglycans are components of normal connective tissue [53]. The ECM and cells can compress the tumor blood vessels, which decreases or inhibits blood flow to the tumor sites, thus resulting in the reduction in oxygen and PS delivery to targeted tissues [83]. Therefore, destroying the ECM might be a useful strategy for enhancing vascularization for an excess supply of O_2_ and ameliorating hypoxia. Studies by Gong et al. [84] investigated the effect of PDT with nanomicelles conjugated to Ce6 (NM-Ce6) when combined with hyaluronidase (HAase), which degrades hyaluronan, a crucial component of ECM in tumors. Gong and colleagues reported that the administration of HAase increased tumor vessel densities and oxygenation of the tumor. The localization of 33 nm-sized NM-Ce6 was investigated after the administration of varying doses of HAase (0, 375, 750, 1500 and 3000 U), and an appreciable increase in tumor uptake with NM-Ce6 was obtained at 1500 U concentration (2-fold) [69]. The significant drop in HIF-1 (Figure 4) following this HAase administration demonstrated an improvement in the tumor’s hypoxic conditions. Furthermore, Gong and colleagues also observed a significant obliteration of 4T1 tumors in balb/c mice after combination treatment with HAase and NM-Ce6 under laser irradiation at 660 nm (2 mW/cm^2^, 1 h). Gong et al. also evaluated the effect of HAase administration on PDT efficacy against lymphatic metastasis, since HAase has the ability to move in drainage lymph nodes, which increased the EPR effect of NM-Ce6 and alleviated hypoxia in both the tumor and metastasis regions.

Liu et al. [85] investigated another dual approach to enhance PDT treatment outcomes: the first injection involved the administration of NPs conjugated with collagenase (CLG) to annihilate tumor ECM by means of the disintegration of collagen, and the second injection consisted of Ce6 modified with liposomes for PDT treatment. The acid-sensitive NPs were composed of Mn^2+^ functionalized with benzoic-imine (BI)-linker and polyethylene glycol (PEG) to form GLG-encapsulated nanoscale coordination polymers (NCPs), denoted as CLG@NCP-PEG NPs. Liu and colleagues reported that CLG@NCP-PEG NPs exhibited an increased bioavailability within the tumor, and therefore CLG was released due to the dissociation of NCPs upon exposure to the acidic TME. The released CLG enzyme resulted in the degradation of collagen, the integral constituent of the ECM, weakening of the ECM structure for enhanced tumor perfusion and alleviation of hypoxia. As a result, the second injection of nanoconjugates, Ce6 functionalized with liposome, demonstrated increased tumor uptake and penetration. Liu et al. concluded that such phenomena, in combination with ameliorated tumor hypoxia, significantly increased PDT therapeutic efficacy in mice.

### 6.5. Inhibition of Tumor O_2_ Consumption

In recent years, great strides have been made to decrease oxygen consumption by counteracting mitochondrial respiration [86]. Several targeting agents can be used to target mitochondria, such as metformin’s or papaverine’s derivatives. These complexes impede mitochondrial activity, which in turn intensifies oxygenation concentration [53]. Studies conducted by Yang et al. [87] developed a photoactive nanocomposite, Mn_3_O_4_@MSNs@IR780, endowed with durable hypoxia mitigation abilities. Mn_3_O_4_ exhibited an increased bioavailability in tumors and interreacted with the H_2_O_2_-enriched TME by disintegrating and catalyzing H_2_O_2_ into O_2_. Subsequently, IR780 was released and photoactivated and spontaneously localized in the mitochondria due to its inherent mitochondrial affinity. Upon exposure to laser irradiation, Mn_3_O_4_@MSNs@IR780 destroyed the mitochondria and reduced cell respiration, which in turn alleviated the pervasive hypoxia of tumor tissue, thus enhancing the therapeutic efficacy of PDT. In vitro studies revealed that the nanocomposite exhibited appreciable mitochondrion-targeted abilities and sustainable inhibition of tumor hypoxia. Furthermore, the greatest photoacoustic signal of HbO_2_ with the lowest Hb was seen in tumors from mice following PDT, demonstrating that these nanoparticles can also mitigate tumor hypoxia in vivo [87]. Song et al. [88] investigated the phototoxicity of liposomes loaded with metformin (Met) (a hypoglycemic agent) and hydrophobic Ce6 (HCe6), denoted as Met-HCe6-liposomes. Met-HCe6-liposomes greatly boosted tumor oxygenation in both in vitro and in vivo-cultured 4T1 cells and yielded ^1^O_2_ under 660 nm laser light. Under normoxic conditions and 660 nm laser irradiation for 10 min, all Ce6 formulations exerted the same phototoxic effects. PDT treatment with Met-HCe6-liposomes or HCe6-liposomes 24 h before 660 nm light irradiation (0.035 W/cm^2^_,_ 30 min) significantly reduced tumor growth compared to free Ce6. The most favorable inhibitory effect was achieved by Met-HCe6-liposome-mediated PDT.

Zuo et al. [89] devised a platelet-based nanocarrier platform to co-deliver W_18_O_49_ NPs and metformin (PM-W_18_O_49_-Met NPs) for PTT and PDT treatment. Platelet membranes shielded W_18_O_49_ from oxidation and immune system barriers and increased the bioavailability of W_18_O_49_ in tumor sites via the EPR effect and enhanced affinity of platelets for cancer cells. PM-W_18_O_49_-Met NPs presented the highest photodamage in in vitro-cultured Raji cells after exposure to 808 nm irradiation (1 W/cm^2^, 10 min) compared to PM-W_18_O_49_ NPs and W_18_O_49_ NPs. Similar observations were noted in in vivo mice bearing Raji-lymphoma xenografts treated with the same compounds and 808 nm irradiation for 10 min. Zou and colleagues concluded that metformin could ameliorate tumor hypoxia by inhibiting oxygen consumption. Hence, PM-W_18_O_49_ NPs greatly increased the production of ROS and heat to allow for simultaneous enhanced PDT and PTT.

A study by Yang et al. [90] constructed a biocompatible nano-PS prepared by means of tamoxifen (TAM) to induce the self-assembly of Ce6 modified with cancer-targeting HA to form the HAS-Ce6/TAM nanocomplex that could dissociate under acidic TME conditions. The presence of TAM and HAS could not hamper the ^1^O_2_ generation ability of Ce6. In vitro-cultured 4T1 cells incubated with HAS-Ce6/TAM for 6 h exhibited a similar phototoxic effect as that observed with free Ce6 or C-HAS-Ce6 NPs when subjected to 660 nm laser irradiation (5 mW/cm^2^, 30 min). In in vivo studies conducted on mice bearing 4T1 tumors, HAS-Ce6/TAM exhibited a prolonged half-life and high bioavailability in tumors, where it underwent rapid pH responsive dissociation to enhance HSA-Ce6 intra-tumoral penetration. The authors noted that HAS-Ce6/TAM NPs induced more phototoxicity on 4T1 tumors than other controls due to the ability of TAM to attenuate hypoxia.

Xia et al. [91] investigated a gelatin-based nanocarrier to promote deep tumor penetration using indocyanine-green bovine serum albumin (ICG-BSA) and atovaquone (ato). This nanoconjugate could dissociate upon exposure to the metallopeptidase 2 enzyme overexpressed by tumor tissue, thereby releasing ICG-BSA and ato in targeted tumors. In HeLa cells, the inhibition of the O_2_ consumption rate by ICG-BSA-ato was comparable to that of ato alone. Both 2 μM of ato and ICG-BSA-ato administered to HeLa cells significantly decreased the O_2_ consumption rate by 50%. A high degree of cytotoxicity was observed in in vitro-cultured HeLa cells treated with ICG-BSA-ato and 808 nm laser irradiation (1 W/cm^2^, 5 min) compared to ICG-BSA. Similar observations were obtained in mice bearing HeLa tumors treated with ICG-BSA-ato, which improved the survival rate to 90% after 3 weeks of PDT treatment. Table 5 summarizes studies that have been conducted to alleviate hypoxia by decreasing tumor O_2_ consumption.

### 6.6. Inhibition of Angiogenic Factors

Hypoxia activates HIF-1 and upregulates VEGF, which results in the formation of blood vessels to supply oxygen and nutrients to tumor tissues [10,53]. Therefore, attenuating the expression of HIF-1, VEGF, and CA-IX during PDT treatment holds great promise for preventing tumor relapse and metastasis by stimulating cell apoptosis over cell survival (Figure 3) [10]. Chen et al. [92] synthesized an anisamide-targeted-calcium-phosphate (LCP) nanocomposite to enhance the target specificity of HIF1α siRNA in SCC4 and SAS cells overexpressing sigma receptors during PDT treatment. HIF1α siRNA nanocomposite effectively suppressed the expression of HIF1α, increased cytotoxicity and significantly inhibited cellular proliferation following the photoactivation of photosan in cultured cells. Similar observations were obtained when HIF1α siRNA nanocomposite was administered to in vivo-cultured SCC4 or SAS. Chen and colleagues noted that photosan-mediated PDT and HIF1α siRNA nanocomposite resulted in a significant 40% reduction in tumor volume after 10 days of treatment [93]. The results obtained from caspase-3, TUNEL, and CD31 marker demonstrated that the combination therapy was more potent than the monotherapy of either photosan-mediated PDT or HIF1α siRNA nanocomposite [93]. Sun et al. [94] developed a novel active-targeted multicomponent cationic porphyrin-grafted lipid (CPGL) microbubble functionalized with HIF1α siRNA, which inhibited the expression of HIF1α. Sun and colleagues noted an efficient release of siHIF upon exposure of in vitro-cultured MDA-MB-231 cells to ultrasound and production of ^1^O_2_ after laser irradiation. Furthermore, an 0siHIF and PDT combined treatment significantly annihilated cancer cells in both in vitro monolayers and xenograft models [95].

Studies by Ferrario et al. [93] investigated the effect of photofrin-mediated PDT treatment on the expression of HIF-1α and VEGF in mouse mammary carcinoma. Antiangiogenic agents were combined with PDT by injecting mice with either an endothelial-activating polypeptide (EMAP-II) to prevent VEGF expression or an angiogenic synthetic dipeptide (IM862) to inhibit the overexpression of HIF-1α and VEGF. Ferrario and colleagues noted a reduced expression of VEGF following PDT treatment combined with antiangiogenic therapy, as well as enhanced phototoxicity of PDT. Studies conducted by Zhou et al. [95] studied the influence of hypericin-mediated PDT in combination with VEGF inhibitors, SU5416 and SU6668. The authors observed a significant regression of nasopharyngeal in vivo tumor modes following PDT treatment combined with SU5416 or SU6668, with a better susceptibility of tumors to PDT with SU6668.

Studies conducted by Broekgaarden et al. [96] co-encapsulated acriflavine (ACF) (a HIF-1 inhibitor) in zinc phthalocyanine (ZnPC) functionalized with endothelium-targeting liposomes (ZnPC-ETLs) to inhibit HIF-1 and increase PDT efficacy in cancer cells. Broekgaarden and colleagues revealed that PDT (671 nm, 500 mW, 15 J/cm^2^) in conjunction with ACF exerted an enhanced phototoxicity on in vitro-cultured human perihilar cholangiocarcinoma SK-ChA-1 cells under normoxic and hypoxic conditions. A similar synergist effect was noted in A431 cells treated with ACF and ZnPC-ETLs-mediated PDT (671 nm, 500 mW, 15 J/cm^2^).

Weiss et al. [97] examined the synergistic effects of an active-targeted visudyne-mediated PDT with an anti-VEGF antibody (bevatizumab) and angiostatic tyrosine kinase inhibitors (TKIs, sunitinib, sorafenib, and axitinib) on two tumor models (A278 human ovarian carcinoma cells and HCT-116 human colorectal cancer cells). A278 and HCT-116 cells were implanted in the chorioallantoic membrane of the chicken emboryo. The most favorable PDT effects (35 mW/cm^2^, 5 J/cm^2^) were achieved with TKIs, particularly sorafenib and axitinib, which attenuated the expression of VEGFR-2 receptors in the tumor’s vasculature. However, bevacizumab showed no synergism with PDT.

Lecaros et al. [98] developed an active-targeted photosan-mediated PDT in conjunction with lipid–calcium–phosphate nanoparticles (LCP NPs) to deliver VEGF-A small interfering RNA (siVEGF-A) to human oral squamous cancer cells (HOSCC), SCC4 and SAS in order to improve PDT efficacy by suppressing VEGF-A expression. Photosan-mediated PDT (640 nm, 320 mW/cm^2^, 100 J/cm^2^, 11 min) in combination with siVEGF significantly reduced tumor volume in SCC4 and SAS models by 70% and 120%, respectively.

Liang et al. [99] designed an active-targeted nanosized (55 ± 2) pH-responsive direct-acting-antiviral (DAA) nanoconjugate functionalized with dimethylanthenone-4-acetic acid (DMXAA) to actively target VEGF receptors and diketopyrrolopyrrole (DPP-4) to disrupt the vasculature region via PDT/PTT processes. Following administration of the nanoconjugate, the acidic TME evoked the release of DMXAA with an increase in ^1^O_2_ generation as well as an efficient photothermal destruction compared to pH of 7.4. In vitro- and in vivo-cultured HeLa cells noted a potent synergistic effect induced by the antivascular activity of DMXAA and PTT/PDT (660 nm, 0.8 W/cm^2^, 4 min) to obliterate tumor cells. Table 6 summarizes studies that have enhanced the efficacy of PDT with angiogenic therapy.

## 7. Active PS Nanocarrier Platforms Incorporated with Anti-Hypoxia Agents for Enhanced PDT Treatment of 3D Tumor Spheroids

Cancer selectivity and sensitivity still pose significant obstacles that could compromise the therapeutic efficacy of anticancer drugs and cause unwanted systemic toxicity [100]. In order to increase cellular internalization and combat hypoxic TME, PS nanocarrier platforms are functionalized with active-targeting biomolecules and anti-hypoxia agents [101]. To date, a number of in vitro and in vivo models have been used and approved to investigate the mechanisms of action, activity, and characteristics of anticancer candidates, including PSs [102]. Even though 2D cell culture models are the most widely used platforms for in vitro testing, 3D cell culture models are receiving a lot of attention because they are more physiologically relevant [15,103,104]. The drawbacks of a conventional “flat” culture, such as cell polarity, the interaction of neighboring cells with the microenvironment, oxygen uptake, and nutrient uptake, may be overcome by creating more realistic conditions [15,105]. Additionally, discovering anticancer drugs using 3D cell culture models may open up new trajectories for enhancing the preclinical testing of drug candidates to improve the new pharmacological approaches, as well as providing helpful data for designing in vivo studies [102,105].

Ma et al. [103] developed an oxygen self-sufficient IR780@O_2_-SFNs/iRGD, which consisted of IR780-loaded pH-sensitive fluorocarbon-functionalized nanoparticles (SFNs) and iRGD as an active tumor targeting peptide that can penetrate deeper within the tumor. SOSG revealed an increased ^1^O_2_ concentration generated by the conjugate in vitro and in vivo upon laser irradiation (808 nm, 2 W/cm^2^ for 20 s). ROS generation measured on 4T1 3D tumor spheroids showed an increased concentration even inside the inner core of the spheroid due to the tumor-penetrating ability of iRGD peptide. The highest degree of cytotoxicity was achieved at 7.5 μg/mL with laser irradiation at 808 nm (2 W/cm^2^ for 20 s), which resulted in a cell viability of less than 20%. A photoacoustic system noted an increased O_2_ concentration in 4T1 solid tumors after injection of the nanoconjugate and irradiation. Furthermore, 4T1 tumors implanted in mice were eradicated within 12 days of treatment by the nanoconjugate and laser irradiation (808 nm, 2 W/cm^2^, 5 min).

Studies by Jung et al. [104] investigated an acetazolamide (AZ) conjugated BODIPY PS (AZ-BPS) to actively target aggressive breast cancer cells (MDA-MB-231) overexpressing carbonic anhydrase IX (CAIX). AZ-BPS showed an enhanced cellular uptake and phototoxicity on in vitro cultured-MDA-MB-231 cells (2D and 3D cultures) compared to BPS, which is an analogous photoactive compound that lacks an acetazolamide unit. Jung and colleagues found that 3D cultures more accurately predicted in vivo outcomes than 2D cell culture models, indicating that they are the best choice for screening both clinically active drugs and preclinical drugs. Additionally, AZ-BPS showed greater in vivo efficacy than BPS in a mouse xenograft model, which was attributed to the inhibition of angiogenesis by both PDT-induced ROS generation and CAIX knockdown [106].

Recent studies by Wu et al. [106] developed an active targeting nanobioconjugate composed of nanoenzyme MnO_2_ colloids and a tumor-homing and -penetrating peptide tLyP-1 chlorin e6 (Ce6) (GO-MnO_2_@tLyP-1/Ce6, denoted as GMtC) to ameliorate hypoxia. GMtC was able to accumulate into the inner core of murine mammary 4T1 tumor spheroids (to a depth of approximately 90 μm) and alleviated hypoxia conditions through the decomposition of H_2_O_2_ to O_2_, which then enhanced the cytotoxic ^1^O_2_ yield under laser irradiation. In vivo dual-modal imaging using magnetic resonance and biofluorescence showed that GMtC accumulated and distributed specifically in tumor areas, thus alleviating the tumor hypoxia. Noticeably, GMtC exerted the highest phototoxic effects (660 nm, 0.5 W/cm^2^, 5 min) on 4T1 tumors without causing any off-target toxicity compared to non-penetrating and no oxygen-generating controls.

To date, very few studies have been conducted in 3D cell culture models using active-target PS nanocarrier systems incorporated with anti-hypoxia agents. Thus, this calls for further research in 3D cell cultures to bridge the gap between in vitro and in vivo studies for improved preclinical phases and successful clinical trial outcomes.

## 8. Conclusions and Perspective

PDT is a minimally invasive modality with increased therapeutic potency and negligible side effects compared to conventional radiation and chemotherapy [107]. However, several lines of evidence indicate that hypoxia is an important hallmark of solid tumors associated with tumorigenesis, tumor progression, metastasis, and failure of PDT treatment [107]. Recent advances in nanotechnology have paved new avenues for approaching the development of novel PDT systems and providing versatile opportunities to address issues encountered by current PDT paradigms. In this review, we highlighted current approaches for circumventing hypoxia limitations in PDT, including (1) increasing oxygen supply, (2) the disruption of tumor ECM, (3) the inhibition of tumor O_2_ consumption and (4) the inhibition of angiogenic factors. Typically, PDT experiments have been performed on traditional 2D monolayer cell cultures. Compelling evidence indicates that less than 5% of preclinical drugs succeed in clinical trials, which causes a significant delay in the development of anticancer drugs [15,17]. One major cause of poor drug response is the inadequacy of the preclinical 2D cell cultures to mimic the human TME [15]. Furthermore, 2D cell cultures lack cellular interactions and an inherent tumor microenvironment that facilitates tumor growth and drug response, which poorly predict the outcomes of clinical trials. Currently, the gold standard for screening anticancer agents is in vivo xenograft models. However, these models are very expensive, time-consuming, and have issues with ethical clearance. Furthermore, they contain non-human host cells that still do not simulate the physiological or biological mechanisms in humans [20]. Three-dimensional cell cultures have received a lot of attention in cancer research because they mimic many aspects of solid tumors and thus serve as an intermediary between 2D cell cultures and in vivo solid tumors. Thus, anticancer effects observed in 3D cell cultures may translate into similar effects in vivo, and hence spheroids may significantly reduce, if not entirely replace, the use of 2D cell culture models in the future. Together with novel anti-hypoxia nanotechnology-mediated PS delivery platforms, 3D in vitro models are expected to significantly reduce the cost of screening preclinical drugs and make anticancer therapies more accessible to the general public. The overall findings of this review demonstrate that very few studies have been performed within 3D cell culture models using active PS nanocarrier platforms with anti-hypoxia capabilities. Thus, this warrants further application of functionalized nanocarriers for targeted PDT treatment and the alleviation of hypoxia by conducting more 3D studies to allow for better screening of preclinical drugs and provide more reliable information for designing in vivo studies.

## Figures and Tables

**Figure 1 ijms-24-02656-f001:**
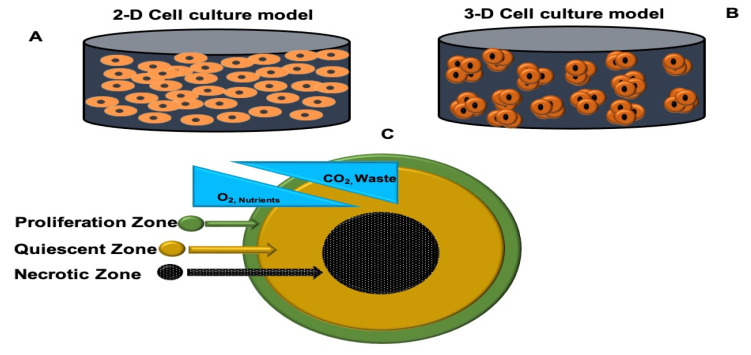
Representation of 2D and 3D cell culture platforms. (**A**) Traditional monolayer cell culture; (**B**) 3D cell culture system; (**C**) distinct layers of a spheroid.

**Figure 2 ijms-24-02656-f002:**
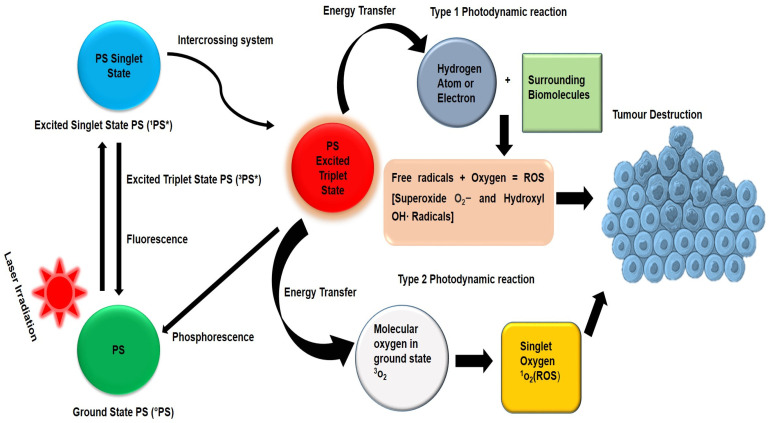
Photodynamic therapy mechanisms of action upon activation of a photosensitizer with a specific wavelength of light.

**Figure 3 ijms-24-02656-f003:**
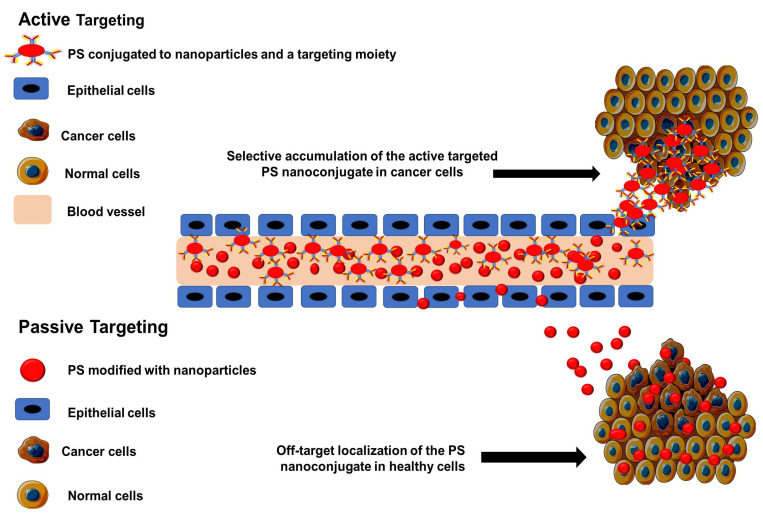
Passive and active nano-PS tumor targeting and delivery systems used for the PDT treatment of cancer.

**Figure 4 ijms-24-02656-f004:**
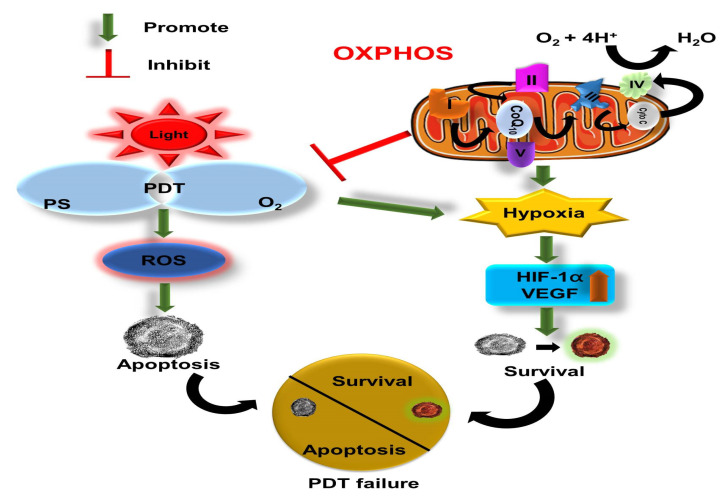
Hypoxia-mediated resistance in PDT treatment.

**Table 1 ijms-24-02656-t001:** An overview of studies that have used HBO for O_2_ supply in PDT.

PS	O_2_ Supplier	Cell Line	Cell Culture Model	Outcomes	Ref
5-ALA	Hyperbaric chamber with 100% O_2_ (21, 100, 200, 300, or 400 kPa)	AY27, WiDr and SW840	2D	PDT was subjected to normoxia-induced lipid peroxidation, which resulted in a significant decrease in cell survival. Additionally, hyperoxic conditions had no effect on lipid peroxidation versus untreated control cells.	[57]
HpD (Photosan) or 5-ALA	HBO (Hyperbaric chamber: 2 atm O_2_)	Esophageal cancer	Clinical trials	PDT/HBO considerably decreased tumor length resulting in a 41.2% survival rate.	[58]
m-THPC	Nicotinamide and carbogen (95% oxygen with 5% carbon dioxide)	H-MESO1	tumor-bearing mice	Nicotinamide therapy and carbogen breathing enhanced tumor oxygenation as well as tumoricidal effects of PDT 24 h post PS administration.	[59]

HBO: hyperbaric oxygen; 5-ALA: 5-aminolevulinic acid; RB: Rose Bengal; HpD: hematoporphyrin derivatives; m-THPC: meta-tetra(hydroxyphenyl)chlorin.

**Table 2 ijms-24-02656-t002:** Summary of studies that have used RBC for O_2_ supply in PDT.

PS	O_2_ Supplier	Cell Line	In Vitro/In Vivo	Outcome	Ref
Ce6	Hb	4T1	2D and in vivo	PDT combined with Hb remarkably alleviated tumor hypoxia and enhanced the efficacy of PDT.	
ZnF16Pc	RBCs	U87MG	2D and in vivo	PDT-RBC exhibited an appreciable tumor inhibitory effect (76,7%) that was attributed to the co-delivery of O_2_ and PS.	[63]
RB	RBCs	U87MG	2D and in vivo	PDT-RBC relieved hypoxia and demonstrated a significant anti-tumor efficacy by remarkably reducing tumor volumes.	[64]
RB	Hb	MCF-7	2D and in vivo	RB loaded with Hb exhibited more severe phototoxicity than RB treatment alone due to more O_2_ and ROS produced during the PDT process.	[65]
TCPP	Hb	4T1	2D	PDT and HB exerted better phototoxicity compared to free PS due to the oxygen supplied by Hb.	[66]
ICG	Hb-RBC	MCF-7	3D tumor spheroids and in vivo	ICG-RBC revealed a remarkable O_2_ self-sufficient PDT effect in 3D tumor spheroids and suppressed HIF-1α expression in vivo.	[67]

Hb: hemoglobin; ICG: indocyanine green; Ce6: chlorin e6; RBC: red blood cells; ZnF16Pc: zinc 1,2,3,4,8,9,10,11,15,16,17,18,22,23,24,25-hexadecafluoro-29H,31H-phthalocyanine; RB: Rose Bengal; TCPP: carboxyphenyl-porphyrin; MB: methylene blue.

**Table 3 ijms-24-02656-t003:** An overview of studies that have used PFCs for O_2_ supply in PDT.

PS	O_2_ Supplier	Cell Line	In Vitro/In Vivo	Outcome	Ref
Fluorous porphyrin	PFC	A375	2D	PFC concurrently delivered O_2_ and PS in cancer cells, which effectively produced more cytotoxic ^1^O_2_ as compared to PS alone.	[70]
ICG	PFH	2D-MDA-MB-231	2D and in vivo	ICG co-loaded with PFH significantly inhibited tumor growth and effectively ameliorated tumor hypoxia.	[71]
ZnF16Pc	PFC	U87MG	2D and in vivo	PS modified with PFC exhibited O_2_ self-enriched PDT treatment, which ultimately obliterated tumor cells without showing any off-target toxicity.	[72]
IR780	PFH	CT26	2D and in vivo	PS loaded into PFH demonstrated significant anticancer effects, while free PS has shown insignificant effects.	[73]

PFH: perfluorohexane; PFC: perfluorocarbon; ICG: indocyanine green; Ce6: chlorin e6; ZnF16Pc: zinc 1,2,3,4,8,9,10,11,15,16,17,18,22,23,24,25-hexadecafluoro-29H,31H-phthalocyanine.

**Table 4 ijms-24-02656-t004:** An overview of studies that used MnO_2_ to decompose H_2_O_2_.

PS	Catalyst	O_2_ Source	Cell Line	Cell Culture Model	Outcome	Ref
Ce6	MnO_2_ nanosheet	H_2_O_2_	B16F10	2D	Ce6 loaded with Mno_2_ nanosheets reacted with H_2_O_2_ and led to massive production of oxygen for enhanced efficacy of oxygen-dependent PDT upon irradiation.	[77]
Ce6	MnO_2_	H_2_O_2_	MB-49	2D and in vivo	MnO_2-_Ce6 nanocomposite significantly enhanced PDT effects by increasing ^1^O_2_ generation by 3.5-fold in vivo and by 2-fold in vitro.	[78]
PPIX	MnO_2_	H_2_O_2_	MCF-7	2D	PPIX and MnO_2_ co-delivery caused more photodamage under hypoxic conditions than when compared to PPIX.	[79]
Methylene blue (MB)	MnO_2_ nanosheet	H_2_O_2_	HeLa	2D	MB-MnO_2_ significantly impeded tumor cell growth versus bare MB due to its ability to produce oxygen for hypoxia amelioration.	[80]
DVDMS	MnO_2_	H_2_O_2_	MCF-7	2D and in vivo	Exhibited a more significant reduction in tumor growth compared to bare DVDMS.	[81]
Ce6	MnO_2_	H_2_O_2_	HeLa	In vivo	Ce6-MnO_2_ demonstrated a significant inhibitory effect on tumor cells compared to unbound Ce6.	[82]

Ce6: chlorin; PPIX: protoporphyrin IX; ICG: indocyanine green; MB: methylene blue; DVDM: sinoporphyrin sodium.

**Table 5 ijms-24-02656-t005:** Summary of studies that reduced O_2_ consumption by modulating TME.

PS	Anti-Hypoxia Agent	Cell Line	Cell Culture Model	Outcome	Ref
HCe6	Metformin (met)	4T1	2D and in vivo	HCe6-met and PDT showed significantly enhanced therapeutic effects versus that of PDT without met.	[88]
W_18_O_49_	Metformin	Raji	2D and in vivo	W_18_O_49_-Met-mediated PDT showed an increased ROS generation, which significantly decreased tumor growth via activation of apoptotic cell death pathway when compared to PDT without met.	[89]
HAS-Ce6-nanocomplex	Tamoxifen	4T1	2D and in vivo	HAS-Ce6 and Tam demonstrated increased half-life and bioavailability, as well as rapid pH-responsive dissociation, to improve Ce6 intertumoral penetration and efficiently attenuate hypoxia for increased PDT efficacy.	[90]
ICG-BSA nanoconjugate	Atovaquone (ato)	Hela	2D and in vivo	ICG-BSA and ato inhibited oxidative phosphorylation, which increased PDT’s therapeutic effects.	[91]

HCe6: hydrophobic chlorin e6; ICG-BSA: indocyanine green-bovine serum albumin; HAS-Ce6: human serum albumin-chlorin e6.

**Table 6 ijms-24-02656-t006:** Overview of studies that inhibited angiogenic factors to enhance PDT efficacy.

PS	Angiogenesis Inhibitor	Cell Line	Cell Culture Model	Outcome	Ref
ZnPc	ACF	A431 and SK-ChA-1	2D	ACF and PDT showed a significant phototoxicity and attenuated the expression of HIF-1 under hypoxic conditions.	[96]
Visudyne	sunitinib, sorafenib and axitinib/bevacizumab	A278 and HCT-116	In vivo	Sunitinib, sorafenib and axitinib significantly enhanced PDT tumoricidal effects and suppressed the expression of VEGFR-2 receptors, while bevacizumab had a negligible effect on PDT.	[97]
Photosan	VEGF-A siRNA	SCC4 and SAS	2D and in vivo	In comparison to the untreated control group, combination therapy significantly reduced the tumor volume in SCC4 and SAS by 70% and 120%, respectively.	[98]
DPP-4	5,6-dimethylxanthenone- 4-acetic acid (DAA)	HeLa and HUVEC	2D and in vivo	PDT combined with DAA significantly destroyed vascular endothelial cells and inhibited tumor proliferation and metastasis.	[99]

ACF: acriflavine; HIF-1α siRNA: hypoxia-inducible factor-1α small interfering RNA; ZnPc: zinc phthalocyanine; VEGF-A: vascular endothelial growth factor-A; DPP-4: diketopyrrolopyrrole.

## Data Availability

Not applicable.

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
