# Peer review of "Anti-Hypoxia Nanoplatforms for Enhanced Photosensitizer Uptake and Photodynamic Therapy Effects in Cancer Cells"

_ijms, 2023, doi:10.3390/ijms24032656_

Round 1
Reviewer 1 Report
Review of Anti-hypoxia Nanocarrier Systems for Enhanced Photodynamic Therapy of Three-dimensional (3-D) Tumour Spheroids
Nkune Williams Nkune and Heidi Abrahamse
Summary: The review paper titled Anti-hypoxia Nanocarrier Systems for Enhanced Photodynamic Therapy of Three-dimensional (3-D) Tumour Spheroids sought to highlight the various methods where studies have previously used nanocarriers containing photosensitizers (PS) for photodynamic therapy (PDT) to combat the issues related to the hypoxic nature of tumor environments. The authors discussed how various studies have addressed the issue of hypoxia by including the following: increasing oxygen supply, improving oxygen circulation, providing an in-situ oxygen source, disrupting the extracellular matrix, and inhibiting angiogenic factors. Their major conclusion was that 2-dimensional (2-D) cell cultures do not accurately reflect the complexity of the tumor environment, which most of their highlighted studies used. Therefore, there is the need to use these nanocarriers in 3-D cell cultures systems to mimic tumor conditions which is more inexpensive and ethically sustainable than using xenograft models.
Feedback:
· Lines 19-21: “… with a particular focus on nanoparticle-based PS delivery systems conducted on 3-D cell cultures” – It does not seem that the studies that the authors highlighted used 3-D cell cultures. I am not sure that this statement is an accurate reflection of the paper.
· Line 21: “PS” – Authors did not introduce the acronym “PS” for photosensitizer (and various acronyms throughout the paper), but they do have an Abbreviations section at the end of the paper.
· Line 49: “ROS” – Again, acronym not introduced.
· Lines 53-55 and 464-466: The authors state the three main strategies to circumvent tumor hypoxia in lines 53-55, which do not quite match the four mentioned in lines 464-466 (conclusion). There is a discrepancy which might be misleading.
· Lines 81-83: I would argue that the authors only investigated “nanoparticle-based PS delivery systems incorporated with anti-hypoxia agents” but did not discuss such systems in 3-D tumor spheroids. The authors only discussed the advantages of 3-D cell culture models in the introduction, then included a few examples in tables (that were not discussed in the text).
· Lines 99-100: “optimal therapeutic window (600-800 nm)” – The authors did not mention why this was the therapeutic window but maybe not necessary (audience is likely those familiar with the NIR window).
· Line 180: “for receptors expressed by cancer cells but not for normal cells” – I think it would be accurate to state “for receptors overexpressed by cancer cells” (which they do state in line 200).
· Figure 3: I understand that the authors want to show that without the targeting moiety, the red dots (PS-NPs) would localize at the tumor area but also around nearby healthy cells, whereas the active targeting (yellow box with arrow) would localize only at the tumor cells, but their depiction is confusing.
· Lines 189-201: Section 4 can be combined with section 3 (or taken out all together), because it is redundant.
· Line 194: “NPs have a large surface to volume” – Is that term accurate? Is it better to state “surface area to volume ratio”?
· Figure 4: Figure is stretched out (cannot read properly). HIF, VEGF, NF-kB, and Bcl-2 where not mentioned in section 5 (starting on line 202), so I think it would be more concise to take these parts out of the figure. HIF and VEGF were mentioned in sections 6.4 and 6.6 but not referenced to this figure.
· Line 234: “5-ALA” – Introduce acronym.
· Line 272: “MCF-7” – No mention of type of cell.
· Line 281: “… denoted (p-Hb-B-NPs) on HepG2 monolayers 620 nm, 25 mW/cm2, 10 min) under normal…” – Minor typo; no closing parenthesis needed.
· Line 282: “N2 atmosphere” – Subscript 2 in N2 for nitrogen
· All tables: The authors only discussed a few of the studies in text but included several references in the tables. I think it is good practice to talk about the references included. For example, in Table 2, the author included reference [50] but did not discuss it, but I would argue that this might be an important one to discuss since it uses a 3-D tumor spheroid model (a part of their argument for this paper).
· Conclusion (starting on line 456): I am not convinced that this paper should include “Three-dimensional (3-D) Tumour Spheroids” in their title. The paper more so covers nanocarrier systems that enhance PDT by addressing the hypoxic nature of tumor environments.
Author Response
REVIEWER 1:
Summary: The review paper titled Anti-hypoxia Nanocarrier Systems for Enhanced Photodynamic Therapy of Three-dimensional (3-D) Tumour Spheroids sought to highlight the various methods where studies have previously used nanocarriers containing photosensitizers (PS) for photodynamic therapy (PDT) to combat the issues related to the hypoxic nature of tumor environments. The authors discussed how various studies have addressed the issue of hypoxia by including the following: increasing oxygen supply, improving oxygen circulation, providing an in-situ oxygen source, disrupting the extracellular matrix, and inhibiting angiogenic factors. Their major conclusion was that 2-dimensional (2-D) cell cultures do not accurately reflect the complexity of the tumor environment, which most of their highlighted studies used. Therefore, there is the need to use these nanocarriers in 3-D cell cultures systems to mimic tumor conditions which is more inexpensive and ethically sustainable than using xenograft models.
Feedback:
- Lines 19-21: “… with a particular focus on nanoparticle-based PS delivery systems conducted on 3-D cell cultures” – It does not seem that the studies that the authors highlighted used 3-D cell cultures. I am not sure that this statement is an accurate reflection of the paper.
Response: Thank you for pointing this out. Lines 19-21 have been edited.
- Line 21: “PS” – Authors did not introduce the acronym “PS” for photosensitizer (and various acronyms throughout the paper), but they do have an Abbreviations section at the end of the paper.
Response: “PS” and other acronyms through have been introduced.
- Line 49: “ROS” – Again, acronym not introduced.
Response: Reactive oxygen species has been added.
- Lines 53-55 and 464-466: The authors state the three main strategies to circumvent tumor hypoxia in lines 53-55, which do not quite match the four mentioned in lines 464-466 (conclusion). There is a discrepancy which might be misleading.
Response: Lines 54-56 have been corrected and now correlate with lines 775-776.
- Lines 81-83: I would argue that the authors only investigated “nanoparticle-based PS delivery systems incorporated with anti-hypoxia agents” but did not discuss such systems in 3-D tumor spheroids. The authors only discussed the advantages of 3-D cell culture models in the introduction, then included a few examples in tables (that were not discussed in the text).
Response: There are very few studies in relation to nanoparticle-based PS delivery systems incorporated with anti-hypoxia agents conducted in 3-D tumor spheroids. However section 6 has been added to support our statement (Line 725-778)
- Lines 99-100: “optimal therapeutic window (600-800 nm)” – The authors did not mention why this was the therapeutic window but maybe not necessary (audience is likely those familiar with the NIR window).
Response. “optimal therapeutic window” has been replaced “by maximum absorption of light at wavelengths from 600 to 800 nm”
- Line 180: “for receptors expressed by cancer cells but not for normal cells” – I think it would be accurate to state “for receptors overexpressed by cancer cells” (which they do state in line 200).
Response: “For receptors overexpressed by cancer cells” was added
- Figure 3: I understand that the authors want to show that without the targeting moiety, the red dots (PS-NPs) would localize at the tumor area but also around nearby healthy cells, whereas the active targeting (yellow box with arrow) would localize only at the tumor cells, but their depiction is confusing.
Response: Thank you for pointing this out. The depiction of active targeting in Figure 3 has been corrected.
- Lines 189-201: Section 4 can be combined with section 3 (or taken out all together), because it is redundant.
Response: Section 4 has been taken out to avoid redundancy.
- Line 194: “NPs have a large surface to volume” – Is that term accurate? Is it better to state “surface area to volume ratio”?
Response: This statement was from the removed section 4. Thank you for the suggest.
- Figure 4: Figure is stretched out (cannot read properly). HIF, VEGF, NF-kB, and Bcl-2 where not mentioned in section 5 (starting on line 202), so I think it would be more concise to take these parts out of the figure. HIF and VEGF were mentioned in sections 6.4 and 6.6 but not referenced to this figure.
Response: Figure 4 has been corrected and NF-kB and Bcl-2 were removed from the Figure. HIF and VEGF were mentioned in sections 6.4 and 6.6 have been referenced to this Figure.
- Line 234: “5-ALA” – Introduce acronym.
Response: 5-aminolevulinic acid added (Line 240).
- Line 272: “MCF-7” – No mention of type of cell.
Response: Type of cell has been mention (breast cancer cells) (Line 307).
- Line 281: “… denoted (p-Hb-B-NPs) on HepG2 monolayers 620 nm, 25 mW/cm2, 10 min) under normal…” – Minor typo; no closing parenthesis needed.
Response: Thank you. Closing parenthesis removed (316).
- Line 282: “N2 atmosphere” – Subscript 2 in N2 for nitrogen
Response: Nitrogen has been subscripted (Line 319).
- All tables: The authors only discussed a few of the studies in text but included several references in the tables. I think it is good practice to talk about the references included. For example, in Table 2, the author included reference [50] but did not discuss it, but I would argue that this might be an important one to discuss since it uses a 3-D tumor spheroid model (a part of their argument for this paper).
Response: The authors are thankful to the reviewer for this suggest. References in all the tables have been discussed.
- Conclusion (starting on line 456): I am not convinced that this paper should include “Three-dimensional (3-D) Tumour Spheroids” in their title. The paper more so covers nanocarrier systems that enhance PDT by addressing the hypoxic nature of tumor environments.
Response: The authors have decided to remove “Three-dimensional (3-D) Tumor Spheroids” from the title since there are very few studies in relation to nanoparticle-based PS delivery systems incorporated with anti-hypoxia agents conducted in 3-D tumor spheroids. The new titles is “Anti-hypoxia Nanocarrier Systems for Enhanced Photodynamic Therapy of Cancer”.
Reviewer 2 Report
This is a review article that tends to cite other review articles and not primary sources. As a result, it is difficult for the reader to decide what is pertinent. As an example, Ref. 36 is cited for a discussion of hydrogen peroxide levels in tumor cells but this is only a review that refers the reader to a paper in Cancer Research (1991). Table 2 in the present manuscript is essentially revision of Table 2 in Ref 36. Ref 35 is claimed to indicate that hypoxia can affect autophagy, drug ‘efflux’ (which presumably means examples of multi-drug resistance that involve outward transport processes) and autophagy but none of this is explained.
With regard to model studies, it is true that monolayers cannot predict for anything but direct anti-tumor efficacy. Moving to more complex systems, e.g., 3D cultures, additional factors can be probed but not immunologic effects or consequences of vascular shutdown. It is stated that ‘the animal models are costly, labour-intensive, controversial with ethical issues, and their outcomes differ depending on animal species.’ All of this is true, but irrelevant to the issue of assessing the efficacy of any proposed PDT protocol. Before regulatory approval will be granted for clinical studies, animal studies will clearly be required. In both 2D and 3D cultures, potassium cyanide will be an effective anti-tumor agent, but not in animal models. Studies in 3D culture may provide additional information on mechanisms of photodynamic efficacy but will never circumvent the need for animal trials.
The citation relating to the ability of autophagy to function as a cell-death mode (Ref 24) is yet another review article and it is not clear where this information comes from. In the context of PDT, autophagy is generally considered to be cytoprotective, being responsible for the ‘shoulder’ on the log dose-response curve. If the authors of this report are going to claim that autophagy is a death mode, they will need to do better than cite a review.
Summary: there are too many citations of other reviews and not enough critical probing of the primary literature. While 3D models may be useful for examining effects not seen in monolayers, animal studies will still be needed to examine systemic effects of photodynamic agents. The authors should try to avoid citing other reviews that may have reached unsupported conclusions, look at the primary literature and see whether they can come up with concepts not noted in prior reviews.
Author Response
REVIEWER 2:
Comments and Suggestions for Authors
This is a review article that tends to cite other review articles and not primary sources. As a result, it is difficult for the reader to decide what is pertinent. As an example, Ref. 36 is cited for a discussion of hydrogen peroxide levels in tumor cells but this is only a review that refers the reader to a paper in Cancer Research (1991). Table 2 in the present manuscript is essentially revision of Table 2 in Ref 36. Ref 35 is claimed to indicate that hypoxia can affect autophagy, drug ‘efflux’ (which presumably means examples of multi-drug resistance that involve outward transport processes) and autophagy but none of this is explained.
Response: Ref. 37 has been added in support of endogenic hydrogen peroxide and all studies in Table 2 have been discussed. Hypoxia-mediated autophagy and multidrug resistance are explained (Line 211-222).
With regard to model studies, it is true that monolayers cannot predict for anything but direct anti-tumor efficacy. Moving to more complex systems, e.g., 3D cultures, additional factors can be probed but not immunologic effects or consequences of vascular shutdown. It is stated that ‘the animal models are costly, labour-intensive, controversial with ethical issues, and their outcomes differ depending on animal species.’ All of this is true, but irrelevant to the issue of assessing the efficacy of any proposed PDT protocol. Before regulatory approval will be granted for clinical studies, animal studies will clearly be required. In both 2D and 3D cultures, potassium cyanide will be an effective anti-tumor agent, but not in animal models. Studies in 3D culture may provide additional information on mechanisms of photodynamic efficacy but will never circumvent the need for animal trials.
Response: The authors are thankful to the reviewer for this explanation. The statement suggesting that 3D cultures could replace animal studies has been removed.
The citation relating to the ability of autophagy to function as a cell-death mode (Ref 24) is yet another review article and it is not clear where this information comes from. In the context of PDT, autophagy is generally considered to be cytoprotective, being responsible for the ‘shoulder’ on the log dose-response curve. If the authors of this report are going to claim that autophagy is a death mode, they will need to do better than cite a review.
Response: Primary sources have been cited to support that autophagy is another mode of cell death induced by PDT (Line 147-158).
Summary: there are too many citations of other reviews and not enough critical probing of the primary literature. While 3D models may be useful for examining effects not seen in monolayers, animal studies will still be needed to examine systemic effects of photodynamic agents. The authors should try to avoid citing other reviews that may have reached unsupported conclusions, look at the primary literature and see whether they can come up with concepts not noted in prior reviews.
Response: The primary literature has been used and the overall findings of this review concluded that very few studies have been conducted in 3-D cell culture models using active-target PS nanocarrier systems incorporated with anti-hypoxia agents. Therefore, this warrants further investigations in 3-D cell cultures to bridge the gap between in vitro and in vivo studies for improved preclinical phases and successful clinical trials outcome (section 6).
Round 2
Reviewer 1 Report
Accepted
Author Response
REVIEWER 1: Comments and Suggestions for Authors
Accepted
Thank you!
Reviewer 2 Report
This review is improved but many issues remain. The title indicates that this is a discussion of nanocarrier systems for promoting efficacy of PDT by circumventing effects of hypoxia. It begins with a general review of PDT that has several off-topic elements, e.g., the nature of accumulation of photosensitizing agents, and a discussion of death and survival pathways. None of this has any direct relationship to the title.
As pointed out in a prior review, this review tends to cite other review articles and not primary sources. Both are looked at from a non-critical perspective. The purpose of such reviews is to aid the reader in deciding what is pertinent, not citing every report regardless of merit.
On line 445, ref 53 is cited to provide evidence that malignant cell types have an elevated hydrogen peroxide level. Ref. 53 refers to a 1991 Cancer Research paper for this information. The latter study involved several tumor lines of human origin, in cell culture. The authors indicate: ‘The generation of large amounts of hydrogen peroxide if it were to occur in vivo, could be important pathophysiologically from several aspects.’ So it is not known whether this might be an artifact of cell culture.
Ref. 82 describes studies involving MnO nanoparticles and MCF-7 cells in vitro . Among the procedures described was the use of the fluorogenic probe DCFDA to examine production of reactive oxygen species (ROS) by photodynamic treatment in vitro. The text reads: ‘After irradiation treatment (0.5 W laser 10 min) or added to H+/H2O2 , cells were promptly washed with PBS and incubated with DCFH-DA and dihydroethidium for 30 min, and intracellular ROS generation was evaluated by flow cytometry and confocal microscopy.’ This is the wrong way to use DCFDA since the ROS formed during irradiation have a half-life in milliseconds. If the probe is not present during irradiation, all that will be detected are some long-persisting lipid peroxides and perhaps hydrogen peroxide. This reference also uses the CCk8 assay to assess survival but this is only a test for the activity of some mitochondrial dehydrogenases. While these are not lethal flaws, they illustrate pitfalls in citing references without examining them.
Ref 68 is described in the text as ‘The mice were subjected to 808 nm laser irradiation (2 W/cm2, 20h).’ To what does 20h refer? In the study, there are two protocols described. In one, the photosensitizing preparation was injected into the tumor, which is not the usual procedure. Irradiation occurred directly after this procedure for 2 consecutive exposures of 10 s each. Another protocol describes iv injection; irradiation occurred 24 hours later.
Ref. 68 attributes the fluorogenic interaction of ROS with DCF-DA to singlet oxygen. Setsukinai et al (J Biol Chem 278, 3170-3175, 2003) reported that DCFDA is 300-fold more sensitive to ●OH than to 1O2.
Minor points: what is ‘supplyrennin’ (title to Table 3). Line 389 implies that hemoglobin is a perfluorocarbon.
Author Response
REVIEWER 2: Comments and Suggestions for Authors
This review is improved but many issues remain. The title indicates that this is a discussion of nanocarrier systems for promoting efficacy of PDT by circumventing effects of hypoxia. It begins with a general review of PDT that has several off-topic elements, e.g., the nature of accumulation of photosensitizing agents, and a discussion of death and survival pathways. None of this has any direct relationship to the title.
As pointed out in a prior review, this review tends to cite other review articles and not primary sources. Both are looked at from a non-critical perspective. The purpose of such reviews is to aid the reader in deciding what is pertinent, not citing every report regardless of merit.
Response: The title has been adjusted to give an accurate reflection of the paper. In a nutshell, hypoxia-responsive nanoplatforms improve the cellular uptake of photosensitizers in cancer cells, attenuate hypoxia-induced survival pathways, and so enhance the overall efficacy of PDT-induced cell death.
On line 445, ref 53 is cited to provide evidence that malignant cell types have an elevated hydrogen peroxide level. Ref. 53 refers to a 1991 Cancer Research paper for this information. The latter study involved several tumor lines of human origin, in cell culture. The authors indicate: ‘The generation of large amounts of hydrogen peroxide if it were to occur in vivo, could be important pathophysiologically from several aspects.’ So it is not known whether this might be an artifact of cell culture.
Response: A primary source, ref 74 has been added to support that hydrogen peroxide levels in cancer cells far exceed that of normal cells.
Ref. 82 describes studies involving MnO nanoparticles and MCF-7 cels in vitro . Among the procedures described was the use of the fluorogenic probe DCFDA to examine production of reactive oxygen species (ROS) by photodynamic treatment in vitro. The text reads: ‘After irradiation treatment (0.5 W laser 10 min) or added to H+/H2O2 , cells were promptly washed with PBS and incubated with DCFH-DA and dihydroethidium for 30 min, and intracellular ROS generation was evaluated by flow cytometry and confocal microscopy.’ This is the wrong way to use DCFDA since the ROS formed during irradiation have a half-life in milliseconds. If the probe is not present during irradiation, all that will be detected are some long-persisting lipid peroxides and perhaps hydrogen peroxide. This reference also uses the CCk8 assay to assess survival but this is only a test for the activity of some mitochondrial dehydrogenases. While these are not lethal flaws, they illustrate pitfalls in citing references without examining them.
Response: Thank you for pointing this out. Ref 82 has been removed due to unrealistic data.
Ref 68 is described in the text as ‘The mice were subjected to 808 nm laser irradiation (2 W/cm2, 20h).’ To what does 20h refer? In the study, there are two protocols described. In one, the photosensitizing preparation was injected into the tumor, which is not the usual procedure. Irradiation occurred directly after this procedure for 2 consecutive exposures of 10 s each. Another protocol describes iv injection; irradiation occurred 24 hours later.
Ref. 68 attributes the fluorogenic interaction of ROS with DCF-DA to singlet oxygen. Setsukinai et al (J Biol Chem 278, 3170-3175, 2003) reported that DCFDA is 300-fold more sensitive to ●OH than to 1O2.
Response: Studies by Ref 68 has been revised (Line 392-409).
Minor points: what is ‘supplyrennin’ (title to Table 3). Line 389 implies that hemoglobin is a perfluorocarbon.
Response: "Supplyrennin" was a typographical error meant for supply. Line 389 implied that perfluorocarbon, like hemoglobin, is an oxygen carrier. The statement has been modified to be more concise.
Round 3
Reviewer 2 Report
This title of this review indicates that the intent is to examine the use of ‘nanoplatforms’ for promoting PDT efficacy by circumventing hypoxia and uptake issues in cancer therapy, The introduction begins by citing the prevalence of cancer, but PDT will have only a very limited role in cancer therapy, Disseminated tumors are not readily treated and there are sites where light delivery is no readily feasible. The Introduction appears to dismiss as ineffective clinical PDT as it is currently practiced although many successful results have been reported. Otherwise, there would have been no regulatory approval of PDT protocols.
It is claimed that irrelevant data from 2D models may not predict for clinical efficacy, but many of the reports cited in this review involve 2D models. While this report is improved, there remain a few issues. In spite of solubility issues (line 116) relatively hydrophobic agents, e.g., BPD, can readily be formulated into a preparation that is readily adapted to clinical PDT. So this is not a major problem in pharmacology. Have any of the techniques proposed in this review been adapted to clinical use?
There are a few trivial errors that should be corrected. A few examples: Line 164 ‘ROS or singlet oxygen’. Singlet oxygen IS an ROS. Line 188: do all malignant cell types overexpress these receptors, or only some? The outward transport process mentioned inline 213 has some specificity and will not act on every photosensitizing agent. Lines 414 and 416, change ‘fluorined’ to fluorinated’. Line 540 and elsewhere: change Mno2 to MnO2. Ref, 77 appears to have some irrelevant material
Section 7 summarizes the report and proposes that more 3D studies are needed. No regulatory agency is going to accept 3D studies as an indication of protocol efficacy. Potassium cyanide is quite effective at eliminating malignant cell types in 3D culture. While 3D cultures can be useful for assessing certain elements of efficacy, every PDT agent in current clinical use was first identified from 2D culture studies. The major issues confronting use of PDT for clinical treatment of cancer are failure of the approach to deal with disseminated cancer, problems with providing adequate irradiation to certain sites, e.g., gliomas, likelihood of hemorrhage if a solid tumor is near a major blood vessel (encountered in early studies where attempts were made to treat large tumors), and lack of a sufficient tumor:normal tissue uptake in some sites, e.g., liver and spleen. It seems unlikely that more 3D studies are going to alleviate any of these problems.
Author Response
REVIEWER 3
This title of this review indicates that the intent is to examine the use of ‘nanoplatforms’ for promoting PDT efficacy by circumventing hypoxia and uptake issues in cancer therapy, The introduction begins by citing the prevalence of cancer, but PDT will have only a very limited role in cancer therapy, Disseminated tumors are not readily treated and there are sites where light delivery is no readily feasible. The Introduction appears to dismiss as ineffective clinical PDT as it is currently practiced although many successful results have been reported. Otherwise, there would have been no regulatory approval of PDT protocols.
Response: Both experimental and clinical studies using first- and second-generation photosensitizers had pointed out the need for developing improved photosensitizers (PSs) for PDT applications and achieving better therapeutic outcome. Despite many positive features, PDT still presents a series of limitations that interfere with its capacity to effectively eradicate cancer. Therefore, the application of nanoparticles in PDT aims to enhance water compatibility of hydrophobic PSs, protect the PS from degradation, increase PS bioavailability, increase tumour selectivity, and permit greater penetration depths for the treatment of deep seated tumours, thus increasing treatment efficacy and reducing side effects.
It is claimed that irrelevant data from 2D models may not predict for clinical efficacy, but many of the reports cited in this review involve 2D models. While this report is improved, there remain a few issues. In spite of solubility issues (line 116) relatively hydrophobic agents, e.g., BPD, can readily be formulated into a preparation that is readily adapted to clinical PDT. So this is not a major problem in pharmacology. Have any of the techniques proposed in this review been adapted to clinical use?
Response: Since traditional 2D models fail to provide native 3D tissue structure, significant deviation has been noticed in transferring the results from 2D culture to in vivo experiments. Therefore, 3-D cell culture platforms may reduce the discrepancy between in vitro studies and in vivo studies. In PDT hydrophobic agents are a major problem because they tend to aggregate under physiological conditions, drastically lowering the quantum yields of ROS production. Furthermore, studies have reported that conventional PSs are digested by immune systems barrier upon entering the body. Thus, nanoparticles will not only improve the solubility of hydrophobic agents but also protect them from biological barriers in vivo since they can mimic biological molecules (line 185-190). The techniques proposed in this review are still in preclinical phases.
There are a few trivial errors that should be corrected. A few examples: Line 164 ‘ROS or singlet oxygen’. Singlet oxygen IS an ROS. Line 188: do all malignant cell types overexpress these receptors, or only some? The outward transport process mentioned inline 213 has some specificity and will not act on every photosensitizing agent. Lines 414 and 416, change ‘fluorined’ to fluorinated’. Line 540 and elsewhere: change Mno2 to MnO2. Ref, 77 appears to have some irrelevant material
Response: Trivial errors have been corrected. Some receptors, like the folate receptor, are overexpressed in several malignancies, whereas other receptors are only cancer cell specific. Therefore, different targeting molecules such as antibodies, peptides, etc. are used to target these receptors depending on the type of cancer. The outward transport process was linked to hypoxia. We have now indicated that it may act on some anticancer agents not on every photosensitizing agent (Line 219). Ref, 77 has been removed.
Section 7 summarizes the report and proposes that more 3D studies are needed. No regulatory agency is going to accept 3D studies as an indication of protocol efficacy. Potassium cyanide is quite effective at eliminating malignant cell types in 3D culture. While 3D cultures can be useful for assessing certain elements of efficacy, every PDT agent in current clinical use was first identified from 2D culture studies. The major issues confronting use of PDT for clinical treatment of cancer are failure of the approach to deal with disseminated cancer, problems with providing adequate irradiation to certain sites, e.g., gliomas, likelihood of hemorrhage if a solid tumor is near a major blood vessel (encountered in early studies where attempts were made to treat large tumors), and lack of a sufficient tumor:normal tissue uptake in some sites, e.g., liver and spleen. It seems unlikely that more 3D studies are going to alleviate any of these problems.
Response: In recent years, researchers have been devoted in developing in vitro 3D culture systems which can emulate human physiological response to drugs and have the potential to capture both efficacy of a drug and potential toxicity in other organs. In the near future, novel 3D techniques such as microfluid devices, body-on-a-chip concept and tissue engineering will allow for the integration of multiple functional units of human organs to mimic whole body physiology and assist researchers in obtaining more reliable results and deepen our understanding of what really happens in vivo. Therefore, the some issues confronting the use of PDT in the clinical setting may be investigated using advanced in vitro 3D systems before moving to in vivo models. Most drug candidates fail clinical trials due to the deceptive data obtained from 2D cell culture models. Thus, 3D studies will not only alleviate this issue but also expedite the lengthy drug discovery processes.